# Task rule and choice are reflected by layer-specific processing in rodent auditory cortical microcircuits

Marina M. Zempeltzi [1✉], Martin Kisse[1], Michael G. K. Brunk[1], Claudia Glemser[1], Sümeyra Aksit[1],
Katrina E. Deane [1], Shivam Maurya[1], Lina Schneider[1], Frank W. Ohl[1,2,3], Matthias Deliano[1] &
Max F. K. Happel[1,3 ✉]

The primary auditory cortex (A1) is an essential, integrative node that encodes the behavioral relevance of acoustic stimuli, predictions, and auditory-guided decision-making. However, the realization of this integration with respect to the cortical microcircuitry is not well understood. Here, we characterize layer-specific, spatiotemporal synaptic population activity with chronic, laminar current source density analysis in Mongolian gerbils (*Meriones unguiculatus*) trained in an auditory decision-making Go/NoGo shuttle-box task. We demonstrate that not only sensory but also task- and choice-related information is represented in the mesoscopic neuronal population code of A1. Based on generalized linear-mixed effect models we found a layer-specific and multiplexed representation of the task rule, action selection, and the animal's behavioral options as accumulating evidence in preparation of correct choices. The findings expand our understanding of how individual layers contribute to the integrative circuit in the sensory cortex in order to code task-relevant information and guide sensory-based decision-making.

[1] Leibniz Institute for Neurobiology, D-39118 Magdeburg, Germany. [2] Institute of Biology, Otto von Guericke University, D-39120 Magdeburg, Germany.
[3] Center for Behavioral Brain Sciences (CBBS), 39106 Magdeburg, Germany. ✉email: mzempelt@lin-magdeburg.de; mhappel@lin-magdeburg.de

A central function of the sensory neocortex is the integration of sensory stimulus features and cognitive aspects in behavioral contexts. However, the underlying integrative circuit mechanisms are still only partially understood. In the case of the auditory system, ample evidence has revealed that the primary auditory cortex (A1) integrates sensory information with other contextual and motor signals and further reflects higher cognitive demands responsible for prediction[1–3], choice accuracy[4,5], and auditory-guided decision-making[6–9]. The salience of behaviorally relevant sounds further critically depends on the exact reinforcement regimes and task rules[10–12], which renders the auditory cortex a multifarious integrative circuit. These and other studies have described corresponding neural correlates on the level of single neuron or population activity recordings.

Some studies have suggested layer-specific differences in the representation of auditory information along the vertical axis of the auditory cortex, with granular layers revealing more accurate tonotopic response properties due to the dominant lemniscal inputs compared to supragranular and infragranular layers[13–15]. In addition, a growing body of human imaging studies has shown attention and task-related modulation of auditory processing in the A1[16–19] based on gross neural or metabolic response measures. However, how the canonical principles of the columnar processing are reflected in the aforementioned multiplexed function of the A1 is very much unknown[9,20].

The objective of the current study was to achieve a deeper understanding of the underlying neuronal observables on a mesoscopic level characterizing the contributions of whole populations of synaptic circuits across the cortical layers. We, therefore, utilized chronic laminar local field potential (LFP) recordings and the analysis of the corresponding current source density (CSD) distribution in the A1 in behaving Mongolian gerbils. First, animals were trained to detect two different pure tone frequencies in a two-compartment Shuttle-box in a "Go/NoGo"-task design. After successful task acquisition, the task rule for one of the Go-stimuli was changed to a "NoGo"-stimulus, so that animals had to discriminate between the two stimuli. Based on single-trial CSD data, we used generalized linear-mixed effect models (GLMM) with a logistic link function in order to effectively predict an animal's behavior. In summary, we found that infragranular layers to be involved in auditory-guided action initiation during tone detection, while activity in supragranular layers, particularly, reflects choice options during tone discrimination. Our study thereby revealed that the relative contribution of cortical layers to the canonical columnar response is modulated by task-dependent features such as the behavioral relevance of the stimulus, its particular contingency and required action, as well as direct decision variables and the choice accuracy. This information is represented as accumulating evidence preceding the animal's choice. Finally, this multiplexed information coded by the layer-specific cortical population activity emphasizes the integrative circuit function of the A1 between bottom-up routed task-relevant sound features and top–down-controlled auditory-guided decision-making.

## Results

**Auditory decision-making during change of the task rule**. We trained Mongolian gerbils in an auditory cued two-way active avoidance shuttle-box task to respond to two pure tones (of frequencies 1 and 4 kHz) presented as conditioned stimuli (CS), while recording LFPs from the primary auditory cortex using laminar multichannel electrodes (Fig. 1). Gerbils were trained in two separate phases. First, in a detection training phase, both CS were assigned with a "Go" contingency and required subjects to change the compartment to actively avoid the unconditioned

stimulus (mild electric foot shock, US). We trained animals over consecutive sessions until they reached a stable detection of both stimuli significantly above chance level (Fig. 1a, b). Thereby, we yielded sufficient data for both behavioral choices. In the subsequent training phase, the contingency of the 4 kHz pure tone was changed to a "NoGo" stimulus (CS−), while 1 kHz was maintained as CS+ (or "Go" stimulus). During this phase, animals needed to discriminate between the two pure tones in order to avoid the US. Here, we classified behavioral choices depending on the response of the animal and the contingency as hit, miss, correct rejection (Corr. Rej.), or false alarm (FA) (Fig. 1a, b). Averaged conditioned response (CR) curves across training sessions for both training phases showed a distinct improvement of task performance (Fig. 1b). During detection training averaged hit rates reach almost 80% for both "Go"-stimuli (1 and 4 kHz). During the initial discrimination phase, CR rates dropped for both stimuli (<10% hit rate). Hence, animals did not transfer the behavioral choice for the 1 kHz pure tone from the detection phase, but completely abandoned their detection-based avoidance strategy. They quickly reassociated the 1 kHz CS with a "Go" contingency and showed increasing hit responses within 1–2 sessions, while FA rates in response to the "NoGo" 4 kHz tone were considerably lower (~10–20%). Over the entire training procedure, the reaction times were found to be mainly after the second CS presented within a trial. Compartment changes started to increase in response to the second CS and were equally distributed over the subsequent 4.5 s of the observation window (Fig. 1c). This suggests that the task design allows the subjects to use at least the presentation of a second CS to evaluate their planned behavioral choice.

**Task rule impacts on columnar sound frequency representation**. Over the entire training, multichannel LFP recordings were obtained by single-shank silicon probes chronically implanted in the primary auditory cortex (Fig. 1d; cf. Supplementary Fig. 2). In an averaged CSD trace, the tone-evoked activity in response to the repetitive CS presentation, reflecting the spatiotemporal feedforward flow of sensory information across cortical layers in the A1[21,22], marked the most prominent laminar response pattern. During detection, we generally observed highly similar CSD patterns in response to the two pure tones (both "Go" stimuli) with respect to the spatiotemporal current flow (Fig. 2a). Initial current sink activity was observed within granular layers III/IV and infragranular layer Vb reflecting cortical depths of main thalamocortical inputs from the ventral medial geniculate body. Subsequent synaptic activity is then routed to supragranular layers I/II and infragranular layers Va and VI. The overall columnar response exceeded the 200 ms duration of the pure tone presentation. In awake, passive listening subjects, CSD profiles were generally similar in response to both pure tones, which is due to considerably similar and flat frequency tuning properties across the entire group of animals measured (Supplementary Fig. 3).

During the discrimination phase, however, the two physically identical stimuli evoked considerably different CSD patterns. While the overall tone-evoked columnar activity in Go-trials showed a marked increase, the activity in NoGo-trials was rather unchanged or slightly decreased (Fig. 2b). In addition, we could reveal that two-dimensional CSD data measured in our experiment generally allows to qualitatively dissociate activity patterns utilizing a support vector machine classifier approach (Supplementary Fig. 4). In order to quantify existing differences of the overall columnar activity strength (Fig. 2b), we compared the root mean square (RMS) values of the AVREC (AVREC RMS; z-normalized (z-norm.)) calculated for the entire trace in each trial

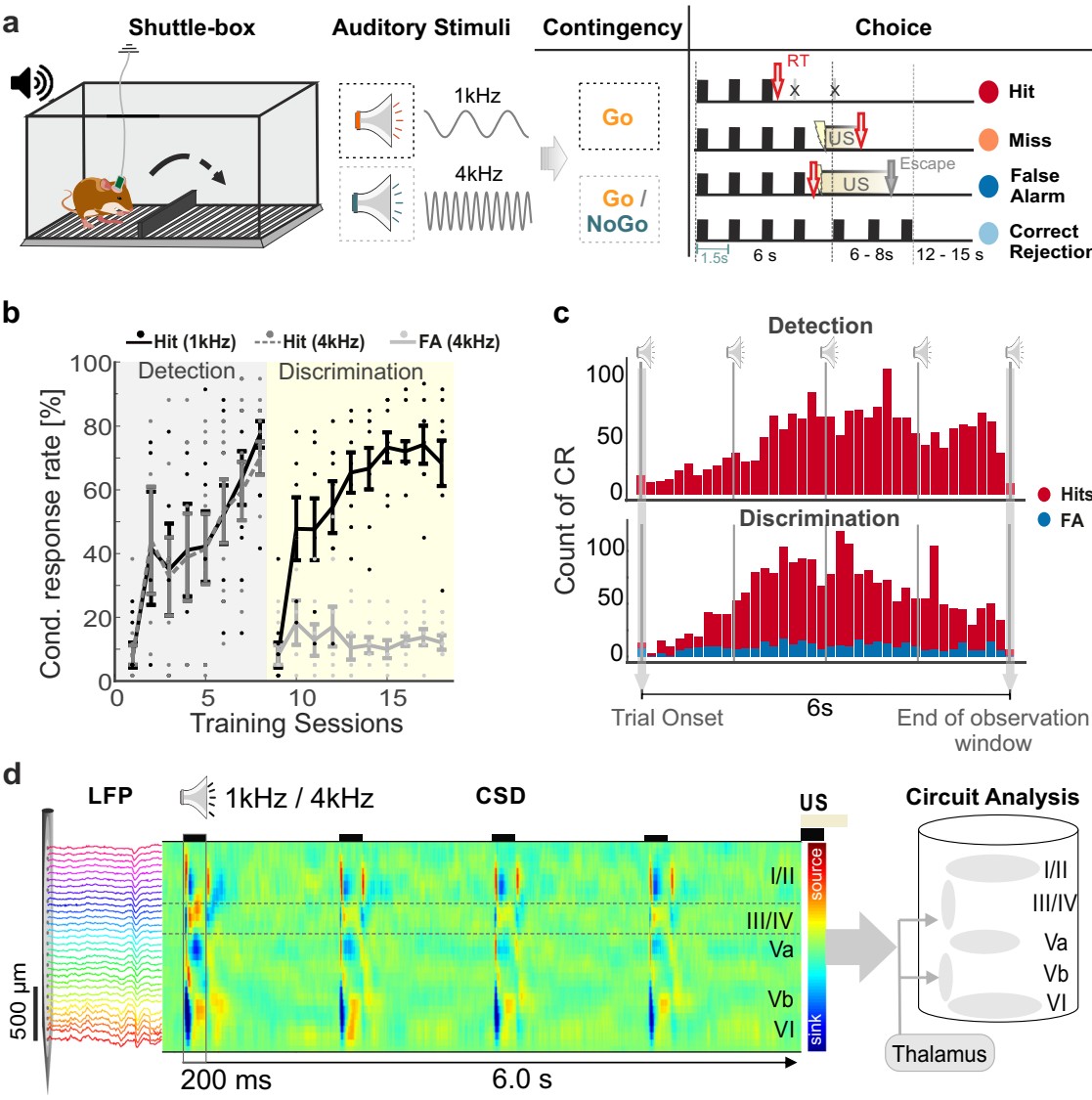

**Fig. 1 Experimental design, learning curves and chronic CSD recording during auditory-based decision-making in a shuttle-box. a** Illustration of the two-way avoidance shuttle-box training with chronic recordings in behaving Mongolian gerbils. Subjects were trained to respond to two different pure tone frequencies (1 and 4 kHz; conditioned stimulus—CS) in a Go/NoGo task design to avoid an unconditioned stimulus (US—mild foot shock). During the discrimination phase the contingency of the CS can be either "Go" (CS+) or "NoGo" (CS−) leading to four possible behavioral outcomes (hit, miss, correct rejection—Corr. Rej., false alarm—FA). Right, Illustration of consecutive CS within a trial, length of the observation window (6 s), interstimulus interval (1.5 s) and behavioral choices. (Gerbil and loudspeaker images taken and modified from https://www.freepik.com/06/2019) (**b**). Averaged conditioned responses (CR) to both CS in the detection and discrimination phase as a function of training sessions (detection/discrimination: $n = 9/8$). During detection (gray area), hit rates reach almost 80% for both "Go"-stimuli (1 and 4 kHz). At the beginning of the discrimination phase (yellow area), conditioned responses dropped for both stimuli (<10% hit rate). The performance gradually increased reaching almost 80% for the hit rates, while false alarm rates stayed around 20%. Error bars indicate the standard error of mean (±s.e.m.). Single dots indicate CR rates of individual subjects. Supplementary Fig. 1 shows corresponding d' learning curves. **c** Histogram with distributions of the averaged CR reaction times over all trials separately for the detection (top) and discrimination (bottom) phase and hits (red) and false alarms (blue). The majority of CR's happen after the second CS presentation. **d** In vivo multichannel LFP recordings were obtained by single-shank silicon probes chronically implanted perpendicular to the surface of the auditory cortex targeting all cortical layers (I–VI). From laminar LFP signals single-trial current source density (CSD) distributions were calculated (here shown is a CSD averaged over 30 repetitions). During CS presentation (200 ms) tone-evoked CSD components appeared as current sink (in blue) and source (in red) activity reflecting the well-known feedforward information flow of sensory information in the A1[21,22]. Supplementary Fig. 2 shows stability of CSD profiles recorded over the training period.

(Fig. 2c). A one-way repeated measures ANOVA (rmANOVA) revealed that during the detection phase the overall activity over the trial between the two CS+ did not differ ($F_{1,8} = 0.20$, $p = 0.668$). During discrimination, the CS+ evoked significantly more cortical overall current flow compared to the CS− ($F_{1,7} = 143.63$, $p < 0.001$). Accordingly, our findings show that the activation

strength of the auditory cortex in response to pure tones depends on the task rule (Fig. 2c, gray insets).

**Auditory cortex represents choice and choice accuracy.** We further differentiated how the cortical recruitment depends on the

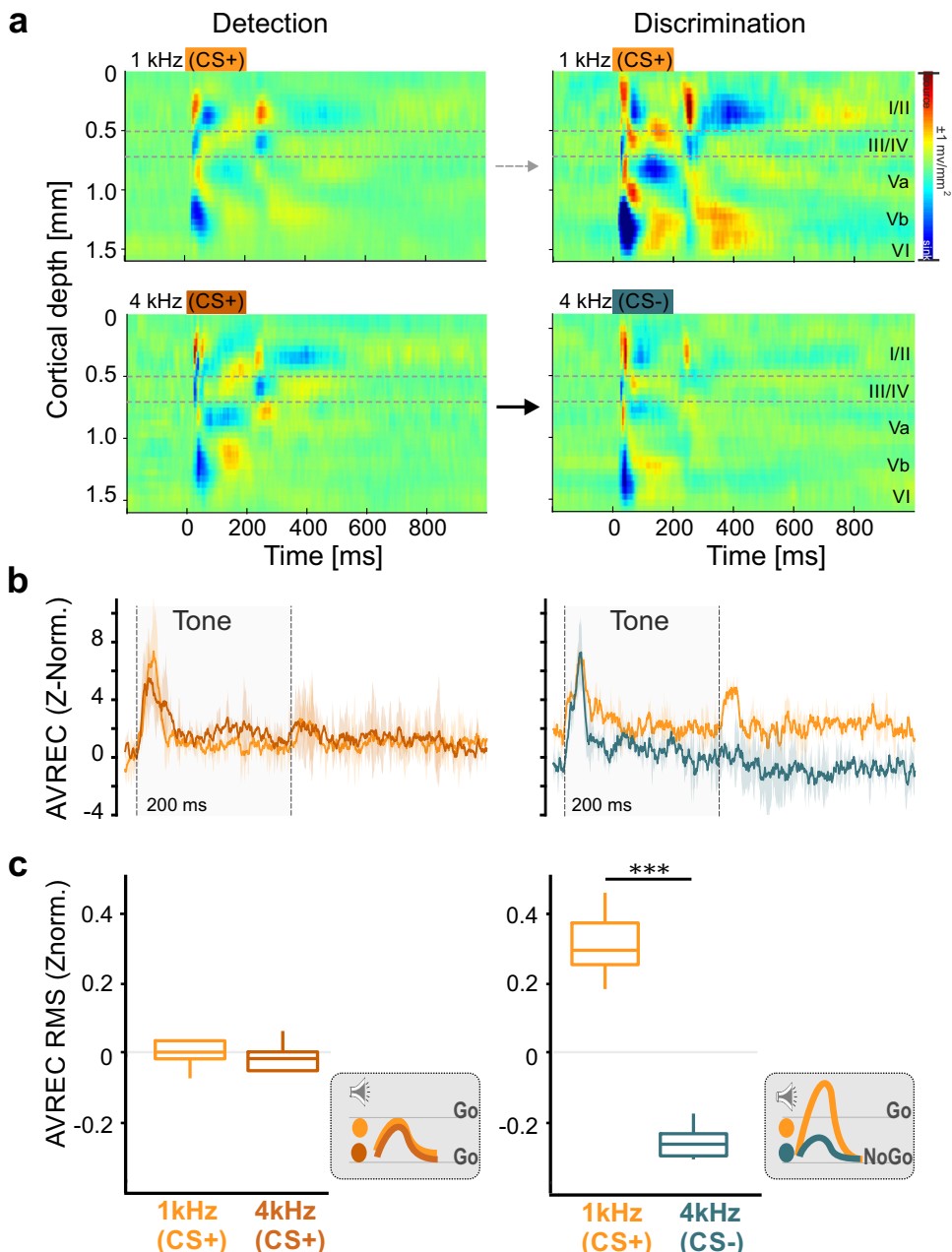

**Fig. 2 Stimulus-related activity during different training phases. a** Representative example of an averaged CSD profile across all trials of the detection (left) and discrimination (right) phase of one subject. The CSD profiles show the tone-evoked activity after the first presentation of both conditioned stimuli within a trial (top: 1 kHz, bottom: 4 kHz; tone duration: 200 ms; indicated by dashed bar in upper left panel). Evoked CSD patterns between the two pure tones frequencies showed no obvious differences during the detection phase but yielded considerably different CSD patterns during discrimination for the CS+. **b** Corresponding raw AVREC traces (z-norm.) for the detection (left) and discrimination (right) phase for the two conditioned stimuli. The shaded error bars indicate the standard error of mean (±s.e.m) of the averaged AVREC traces. The raw traces between the two pure tones frequencies showed no obvious differences during the detection phase, but considerably different activity between CS+ and CS− trials during discrimination. **c** RMS values of the AVREC (time window of 500 ms beginning at each tone presentation and z-normalized) shown for each of the four consecutive CS and separated by the different behavioral outcomes during the two task phases (detection/discrimination: $n = 9/8$). Box plots represent median (bar) and interquartile range, and bars represent full range of data. Significance bar indicate differences revealed by pairwise testing (one-way rmANOVA; $p < 0.05$). Schematic illustration of the evoked cortical activity in dependence of stimulus frequency and task rule are shown in gray inserts.

decision made by the animal. We compared z-norm. AVREC RMS values during a 500 ms window beginning with each CS onset. In a first step, we tested the dependence of the animals' choice options and the time bins throughout the trial, by a two-way rmANOVA with the main factors outcome and tone order (Fig. 3). We found significant effects for both main factors, outcome and tone order, as well as their interaction in both the

detection and discrimination phase (Detection: outcome $F_{3,24} = 40.63$ ($p < 0.001$, $\eta_{gen}^2 = 0.66$), tone order $F_{3,96} = 26.58$ ($p < 0.001$, $\eta_{gen}^2 = 0.25$), interaction $F_{9,96} = 10.05$ ($p < 0.001$, $\eta_{gen}^2 = 0.27$); Discrimination: outcome $F_{3,21} = 31.67$ ($p < 0.001$, $\eta_{gen}^2 = 0.68$), tone order $F_{3,84} = 26.39$ ($p < 0.001$, $\eta_{gen}^2 = 0.23$), interaction $F_{9,84} = 17.52$ ($p < 0.001$, $\eta_{gen}^2 = 0.38$), Supplementary Table 1a).

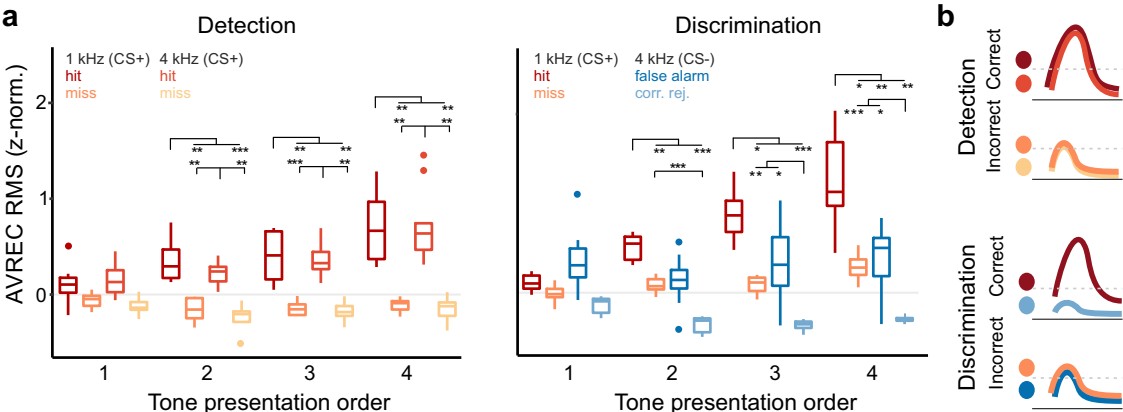

**Fig. 3 Behavioral choices and contingency are both reflected in population activity of the A1.** Averaged AVREC RMS values (500 ms window at CS onsets) plotted with respect to the conditioned stimuli and behavioral choice. **a** Left, During the detection phase evoked activity was significantly higher during hit trials compared to miss trials independent of the stimulation frequency (detection/discrimination: $n = 9/8$). Right, In the discrimination phase, cortical activity was strongest during correct hit trials and lowest during correct rejections. During trials of incorrect behavioral choices (miss/false alarm) tone-evoked activity was characterized by intermediate amplitudes and did not differ. Box plots represent median (bar) and interquartile range, and bars represent full range of data. Dots represent outliers. Significance bars indicate differences revealed by a two-way rmANOVA and corresponding posthoc tests with Holm-corrected levels of significance (see Supplementary Table 1) (**b**). In summary, cortical activity was generally higher in trials in which animals showed a conditioned response in comparison to trials where animals stayed in the compartment. Cortical activity differed strongest between correct behavioral choices, namely hits and correct rejections.

In order to test the differences of the behavioral outcomes at each time point separately, we used restricted Holm-corrected posthoc comparisons at each of the CS+ presentations. During detection training, posthoc tests revealed that the evoked AVREC RMS values (z-norm.) after the first CS presentation are similar for hit and miss trials. Consecutive CS+ presentations evoked significantly higher RMS values during hit trials compared to miss trials (Fig. 3a). These findings were independent of the actual stimulation frequency (1 or 4 kHz). In the discrimination phase, we found a significantly different recruitment of auditory cortex columnar activity depending on frequency and choice of the subjects at the second and ongoing CS presentations throughout a trial (Fig. 3a). Cortical activation in the 500 ms time window around the first CS showed only minor differences. During later CS presentations, we found a stable pattern of columnar activity. During hit trials, cortical activation was significantly highest compared to all other classes. Correct rejections showed the lowest cortical recruitment. In contrast, cortical activation during miss and FA trials did not differ at any CS presentation throughout the trial. Note that cortical recruitment was generally stronger during trials in which animals reported a compartment change (hits > misses; FAs > correct rejections), comparable to findings in the detection phase. However, as the cortical activity during miss and FA trials did not differ significantly, the variability of cortical activation in our data cannot be explained by a mere correlate of motor responses or motor planning, but also depended on the contingency of a stimulus. Indeed, the strongest difference observed was between the two correct choice options of the animal, namely between hits and correct rejections. Hence, cortical recruitment during detection was influenced to a larger degree by the behavioral action taken by the animal, rather than the physical stimulus characteristic of tone frequency. During discrimination, cortical recruitment was influenced by the frequency, coding for the contingency of the stimulus, and the choice accuracy of the taken action (Fig. 3b). In addition, we analyzed a time window of 500–1000 ms after each stimulus presentation (stimulus duration: 200 ms), in order to separate the relative modulation of cortical layer activity by sensory-driven effects from the task-related, but potentially temporally distributed information. This analysis revealed a similar pattern of

cortical activity being modulated by the choice accuracy, which is hence present also independently from the stimulus-dominated auditory response (Supplementary Fig. 5).

**Layer- and task rule-specific representation of contingency.** In order to investigate the contribution of cortical layers to the observed effects, we analyzed binary classes on a single-trial level using GLMM[23]. The GLMM analysis revealed that in the detection phase, the AVREC trace RMS (z-norm.) was not dependent on the presented frequency of the two CS ($R^2$m = 0, ns.; Fig. 4a). During the discrimination phase, an increase in the AVREC trace RMS was a reliable predictor that the 1 kHz "Go" stimulus was presented ($R^2$m = 0.17, $p < 0.001$; Fig. 4a). Hence, the columnar activity in auditory cortex in response to the same CS differed in dependence of the task and was only separable when both had contrasting contingencies. We further applied the GLMM to the RMS value measured over the entire trace activity within single-cortical layers (I/II, III/IV, Va, Vb, and VI) in order to reveal the source of the aforementioned results on a layer-specific level (Fig. 4b). In the detection phase, the two CS+ used as binary class in the GLMM could not predict significantly for any particular cortical layer. When we applied the GLMM for the two CS during the discrimination phase, now reflecting the distinct contingency of CS+ and CS−, we observed a moderate prediction of the model with an $R^2$m of 0.12 for only the granular input layers. Detailed results for each model are reported in Supplementary Table 2.

Next, we used the GLMM to predict the behavioral choices rather than the stimulus frequency (Fig. 5a). Therefore, we compared the AVREC RMS values (z-norm.) of 500 ms windows beginning at the onset of the tone presentation that preceded an active avoidance response of the animal (hit/FA) or around the last CS in the observation window in trials without a CR (miss/Corr. Rej.). During the detection phase, a higher AVREC RMS was a robust predictor for trials with a correct hit response compared to miss trials with lower overall cortical activity ($R^2$m = 0.32, $p < 0.001$; Fig. 5a). In order to test the contribution of distinct cortical layers to the coding of different behavioral choices, GLMM predictions were calculated for RMS values of each layer separately (Fig. 5b). We found that during the

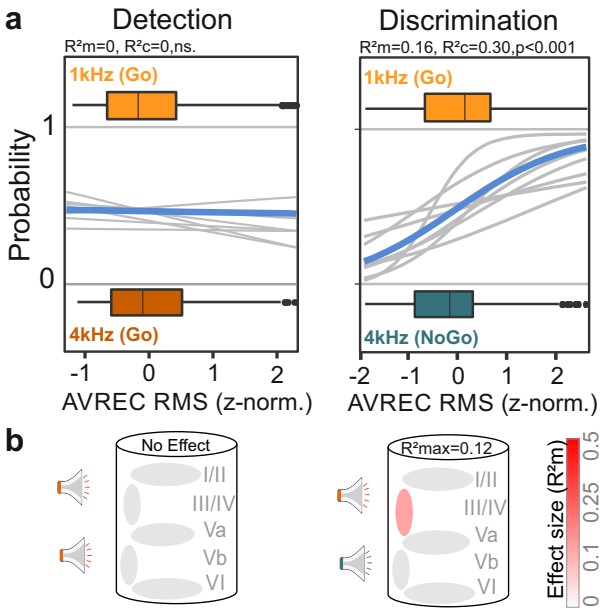

**Fig. 4 Representation of contingency, not frequency revealed in synaptic population activity of granular input layers.** Parameters of interest were analyzed on a single-trial level using generalized linear-mixed effect models (**a**). Logistic regression curves show the probabilities of the presented CS (1 and 4 kHz as the dependent variable) for individual subjects (gray) and as an average (blue). The box plots above and below the curves represent the mean (bar), interquartile range (box), and the full range of data (whiskers). The AVREC trace RMS did not predict the frequency of the conditioned stimuli (1 and 4 kHz) during the detection phase (left, $R^2m = 0$, $R^2c = 0$, ns.). During discrimination an increase in the AVREC trace RMS significantly indicated that the 1 kHz "Go" stimulus was played ($R^2m = 0.16$, $R^2c = 0.30$; $p < 0.001$). Hence, auditory cortical activity in response to the same conditioned stimuli differed in dependence of the task. **b** GLMM's were applied to RMS values measured within single-cortical layers (I/II, III/IV, Va, Vb, and VI). The illustration of the cortical column below indicates the GLMM predictability based on data from corresponding layers to the binary behavioral choice combinations. The color illustrates the effect size for the model-based $R^2m$ (gray = no effect to red = strong effect). The top $R^2m$ value ($R^2max$) depicts the best fit result for all layers tested. In the detection phase, the two CS+ used as binary class in the GLMM revealed no significant prediction for any particular cortical layer. During the discrimination phase we observed a moderate prediction of the model with $R^2m = 0.12$ for the granular input layers. For detailed results of each GLMM see Supplementary Table 2.

detection phase, cortical activity in infragranular layers were good predictors ($R^2m = 0.2–0.25$, $p < 0.001$), while supragranular and granular layer were less accurate ($R^2m = 0.11–0.19$, $p < 0.001$; Fig. 5b; cf. Supplementary Table 3).

During the discrimination phase the AVREC RMS predicted the choice outcome during "Go"-trials (hits vs. misses) with a moderate effect size ($R^2m = 0.18$, $R^2c = 0.38$, $p < 0.001$; Fig. 5a). For "NoGo"-trials the GLMM was able to predict the outcome with a high effect size: FAs were effectively predicted by stronger cortical recruitment than measured during correct rejections ($R^2m = 0.27$, $p < 0.001$; Fig. 5a). During discrimination, granular and supragranular layers appear to be important for the differential representation of the behavioral choice in "Go"-trials ($R^2m = 0.14–0.18$; $p < 0.001$ Fig. 5b). During "NoGo"-trials, the RMS value of all cortical layers except of layer VI were good predictors for the trial outcome ($R^2m = 0.10–0.17$, $p < 0.001$), while supragranular layers were also the best predictor between FAs and correct rejections (Fig. 5b; see Supplementary Table 3).

**Choice accuracy is represented throughout cortical layers.** We compared correct and incorrect choice options of the subjects showing that only correct choices lead to a distinguishable cortical circuit activation (Fig. 6). The AVREC RMS could predict the outcome in correct trials with a high effect size: Correct "hit" responses can be predicted by higher RMS values of the AVREC trace in the time window before the actual decision compared to the time window at the trial end during correct rejections ($R^2m = 0.45$, $p < 0.001$; Fig. 6a). In contrast, the two incorrect choices "FAs" and "miss" were not predictable by the GLMM ($R^2m = 0.04$; n.s.; Fig. 6a). The layer-specific analysis further revealed that particularly the supragranular layer activity contributed to the differential cortical activation between the correct choice classes ($R^2m = 0.18–0.51$; $p < 0.001$; cf. Supplementary Table 4; Fig. 6b). Nevertheless, all cortical layers were recruited in a distinct way, so that the whole cortical column differs in activity during hit and Corr. Rej. choices. In accordance with the insignificant GLMM result on the overall columnar activity measured by the AVREC RMS, also no cortical layer activity could predict the two incorrect choices (FA/miss; Fig. 6b).

**Laminar accumulating evidence of task-related information.** In order to reveal the temporal accumulation of such task-dependent information during a trial, we further analyzed time-resolved $R^2m$ values for behavioral choices, which generally increased over the trial duration preceding an animal's reaction (Fig. 7). While infragranular layers Va–VI showed moderate $R^2m$ values even 2–3 CS+ presentations before the actual response (up to 3–4.5 s), layers III/IV and I/II only allowed moderate predictions of the animal's choice. During discrimination, activity in cortical layers I/II and III/IV, however, allowed us to correctly predict the occurrence of correct rejections up to three CS presentations before the animal's reaction, which is particularly pronounced in contrast to FA trials. The largest effects were found when comparing hit vs. correct rejections, revealing the accumulating evidence of choice accuracy over the trial duration.

## Discussion

In this study, we chronically recorded LFPs and calculated CSD distributions from the primary auditory cortex of behaving Mongolian gerbils. We use auditory instrumental conditioning in two consecutive training phases in a Go/NoGo shuttle-box task with detection and consecutive discrimination of two pure tone frequencies centered within the optimal hearing range of the gerbil. After successful detection, animals had to abandon their initially learned strategy and "reassociate" one of the two CS with a new meaning during the discrimination phase. We found that such a switch of the task rule caused the animals to completely abandon the previous but still valuable "knowledge" about parts of the stimulus representation, i.e. the CS which kept the same meaning. Hence, they had to relearn a new set of behavioral action-outcome contingencies (Fig. 1b).

Based on the laminar distribution of CSDs, we demonstrate that not only sensory but also task- and choice-related information is represented in the neuronal population activity distributed across cortical layers. The frequencies of two pure tones used as CS were only differentially represented in the A1 when they differed in their contingency, i.e. when their discrimination was behaviorally relevant for the task. Cortical activity also differed with action selection, generally showing a higher recruitment during trials where the animal initiated a compartment change. During the detection phase, infragranular layers contributed most to those differences. In contrast, recruitment of synaptic activity in supragranular layers was the most robust predictor for choice outcomes during the discrimination phase (Fig. 5). We further

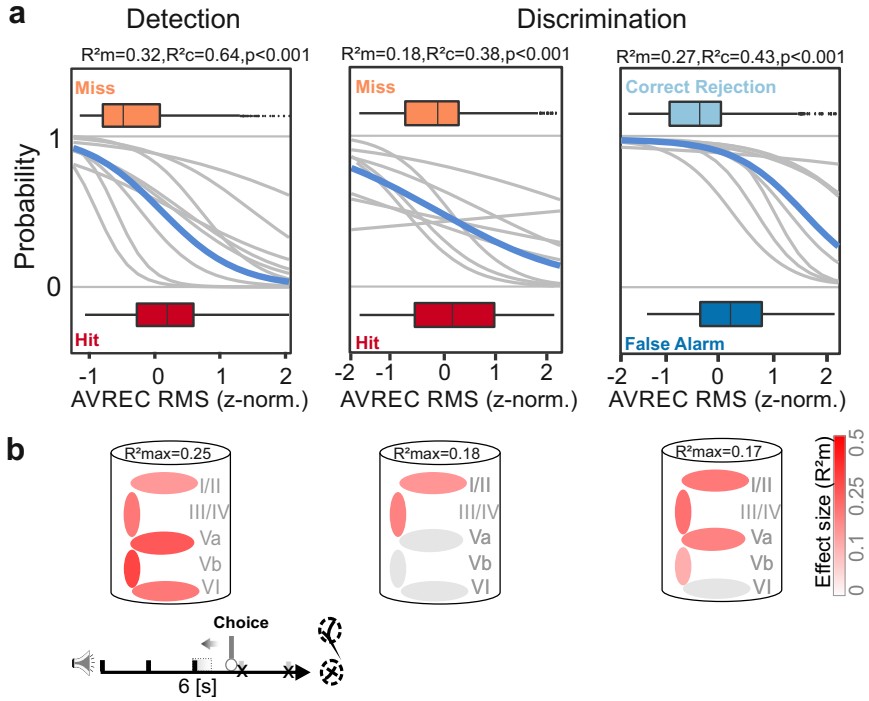

**Fig. 5 Layer-specific contribution to behavioral choice. a** GLMM and logistic regression analysis was used to predict the behavioral choice of the subjects. Left, During the detection phase RMS values of the AVREC (z-norm.) in the 500 ms time window around the CS presentation which was initiating a hit response was significantly higher compared to the fourth CS during miss trials ($R^2$m = 0.32, $p < 0.001$). Middle, This was also true for the discrimination phase, although with a more moderate effect size ($R^2$m = 0.18, $p < 0.001$). When comparing data from "NoGo" trials, false alarm and correct rejections could be predicted with a high effect size ($R^2$m = 0.27; $p < 0.001$). The box plots above and below the curves represent the mean (bar), interquartile range (box) and the full range of data (whiskers). **b** The illustration of the cortical column below indicates the GLMM predictability based on data from corresponding layers to the binary behavioral choice combinations. The color illustrates the effect size for the model-based $R^2$m (gray/red scale). GLMM predictions for each layer showed that cortical activity from all layers were moderate to good predictors ($R^2$m = 0.1–0.25; $p < 0.001$). Higher effect sizes were observed particularly at deeper layers Va, Vb, and VI, for the two possible choices (hit/miss). This finding was independent of the actual spectral content of the presented stimulus (1 kHz/4 kHz; see Fig. 3). During the discrimination phase, granular and supragranular layers appear to be important for the differential representation of the behavioral choice in "Go"-trials ($R^2$m = 0.14–0.18; $p < 0.001$). For "NoGo"-trials, the GLMM revealed that false alarms are accompanied by significantly higher activity in all cortical layers except of layer VI compared to correct rejections ($R^2$m = 0.17; $p < 0.001$). Supragranular layers were also the best predictor between false alarms and correct rejections classes. For detailed results of each GLMM see Supplementary Table 3.

found a robust representation of the choice accuracy independent of the actual action selection of the animal. While all cortical layers showed a stronger recruitment during correct hit trials compared to correct rejections, we did not observe differences between FAs and misses (Fig. 6). Time-resolved analysis of cortical activity allowed to measure these representations to emerge over the trial duration time in agreement with the concept of accumulating evidence in sensory cortex. Hence, our findings argue for a multiplexed representation of stimulus- and task-related features distributed across cortical layers. Of importance, plastic shifts of auditory tuning properties would not explain our findings (Supplementary Fig. 3), strongly arguing for a response modulation in the auditory cortex by the task-relevant contingency, sensorimotor planning, and decision-making.

Specifically, a GLMM analysis revealed that during discrimination, stimulus-dependent features between Go- and NoGo-stimuli are represented differentially as the sound frequency attained a behavioral relevance due to the shift of the task rule. During the initial detection task, activity in all cortical layers, however, reflected the initiation of an active avoidance response. The task-irrelevant sound frequency was not differentially represented on a columnar response level. After switching to the more demanding discrimination task employing the same pure tone stimuli, synaptic circuits within mainly granular input layers and supragranular layers reflected the behaviorally observed

discriminability between the stimulus classes "Go" and "NoGo." Hence, the task structure affected the columnar representation of auditory information to otherwise identical pure tones.

These task-dependent representations emerge as accumulating evidence throughout the trial and are most strongly represented right before a behavioral choice of the animal (see Figs. 3 and 7). During the discrimination phase, animals needed to differentially represent the sound frequency of the two CS to successfully perform the task. Henceforth, the need of spectral integration was likely to be behaviorally more important during this phase. Here, we found that thalamic input layers III/IV and supragranular layers I/II were particularly strongly recruited during trials that led to an active CR. Activity during hits and FAs was higher compared to misses and correct rejections, respectively. This might reflect the need for more cross-columnar communication within supragranular layers in order to integrate the spectral content of a presented conditioned stimulus, and its behavioral relevance to promote the correct behavioral choice[22,24–26]. Accordingly, cortical activity was modulated in supragranular layers well before a behavioral choice was made (Fig. 7).

Therefore, we propose that the representation of stimulus features in sensory cortex, such as tone frequency in the A1, does not depend alone on the transmission process of the sensory information via the primary sensory pathways, but is substantially modulated by the behavioral need and the behavioral relevance of

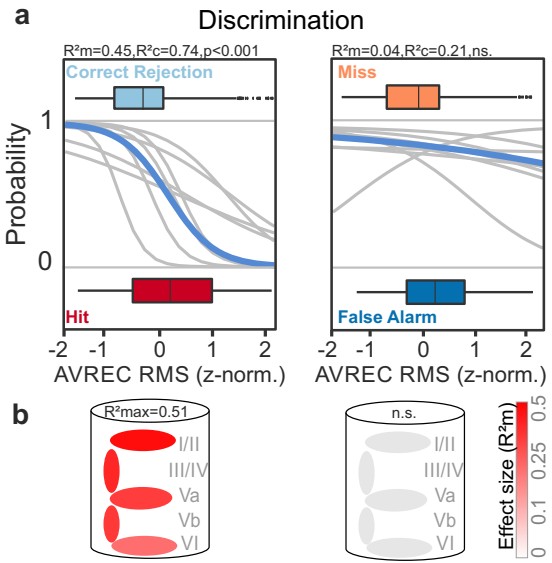

**Fig 6 Representation of choice accuracy across layer-specific population activity in A1. a** Predictability of correct (left) and incorrect (right) choices during the discrimination phase were modeled by GLMM and logistic regression. Correct "hit" responses can be very robustly predicted by higher RMS values of the AVREC trace in the time window before the actual decision compared to the time window at the trial end during correct rejection responses ($R^2$m = 0.45; $p < 0.001$). In contrast, the two incorrect choices "false alarms" and "miss" were not predictable by the GLMM ($R^2$m = 0.04; n.s.). The box plots above and below the curves represent the mean (bar), interquartile range (box) and the full range of data (whiskers). **b** The illustration of the cortical column below indicates the GLMM predictability based on data from corresponding layers to the binary behavioral choice combinations. The color illustrates the effect size for the model-based $R^2$m (gray/red scale). Activity from all cortical layers contributed to the differential cortical activation between the correct choice classes, while the largest effect size was found for supragranular layers ($R^2$m = 0.51; $p < 0.001$). In accordance with the insignificant GLMM result on the overall columnar activity measured by the AVREC, also no cortical layer activity could predict the two incorrect choices (false alarm/miss). For detailed results of each GLMM see Supplementary Table 4.

a stimulus. Such influence is based on the recurrent circuitry between auditory cortex and higher order top–down regions, e.g. parietal and frontal areas[4,27–32]. This might reflect a neuronal basis for auditory response properties of frontal lobe neurons[33], its involvement in auditory detection and discrimination[34,35] and fast top–down response modulation of A1 neurons during behavior via the frontal cortex[36].

Another important topic are correlates of movement responses in sensory cortex. Motor initiation has been reported before to enhance or suppress sensory-driven activity in other (primary) sensory cortices depending on region, system, and task-engagement[30,37]. From our data we hypothesize that, during the detection, the tone-evoked activity in the primary auditory cortex may be modulated by auditory-guided motor initiation[11,38,39]. The distinct sound frequency of a pure tone seems less determining on the activity strength. Deep output layer activity (layers Va–VI) showed a significant increase of activity during hit trials. This is in accordance with the findings that neurons in these layers convey information to downstream motor centers, as the basal ganglia or the striatum, which play an important role for the control of motor decisions by the sensory cortex[40,41]. Further, the selection of an appropriate action might also be conveyed directly to motor cortex via direct anatomical projections[11,42]. Ample evidence argues that our findings reflect a motor-related

modulation of the cortical physiology, rather than a movement artifact. In our data, auditory cortex activity reflected the initiation of motor actions during detection learning most prominently in deeper layers. Hence, motor-related signals were reflected on a layer-specific level while showing a conserved spatiotemporal profile of the tone-evoked CSD, which is in strong favor of a motor-related modulation of the cortical physiology. A muscle correlate, as a far-field artifact, would have affected all recording channels. We controlled this by a trial-by-trial analysis excluding such trials (Fig. 2a). Here, the reference-free CSD measurement might be an effective filter. During discrimination, the cortical activity was less accurate in predicting motor response initiation but was more accurate in predicting correct choice options (Figs. 3 and 6). Cortical population activity did not differ during FA and miss trials. However, cortical activity was elevated between the consecutively presented CS during hit trials. This argues for an accumulative evidence about the stimulus contingency that the animals kept persistently over the trial, which was instructive for an auditory-guided action (Fig. 7). Differences between hit and FAs argue that the motor-related preparatory signal cannot fully explain the variability in our data set. Rather, we find a combinatorial representation of stimulus contingency, task rule, selection accuracy, and motor initiation that accumulates in its richness over the duration of a trial until the actual decision (Fig. 7).

The here described modulation of the cortical activity by contingency and motor initiation reflects a cortical correlate of choice accuracy: in the discriminant Go/NoGo-paradigm, we found all cortical layers to be more strongly activated during correct hits compared to correct rejections (Figs. 6 and 7). In contrast, cortical activity during FAs and misses did not differ. Hence, the cortical representation of spectral information during discrimination training (see Fig. 3) in our experiment was dependent on the accuracy of the promoted behavior. Several studies also observed an enhanced representation of target stimuli that initiated an auditory-guided motor response in various Go/NoGo discrimination tasks[10,43,44]. Studies in rats and humans suggested that lower activity during correct rejections might reflect an active inhibition of motor or cognitive responses during a Go/NoGo task[45,46]. Others, however, found higher cortical recruitment during correct rejections compared to hit trials in a Go/NoGo task in the macaque A1[11]. More research is needed to link the potentially inherent neuronal variability to the exact task design at hand, aversive or appetitive reinforcing regimes, and stimulus characteristics which may partially explain contradictory findings[12,47].

Another relevant aspect is the temporal relation of the observed effects to the repetitive tone presentation throughout the trial in our task design. The time windows of 500 ms around the consecutively presented CS covered the sensory-dominated columnar response (Fig. 2a). Other reports of choice-related activity in the auditory cortex during discrimination of tone events also reported that such representation accumulates until the animal's decision[5,48]. We further found a comparable modulation of cortical activity patterns present in a time window of 500–1000 ms after each stimulus presentation in order to separate the relative modulation of cortical layer activity by sensory-driven effects from the task-related, but potentially temporally distributed, information. Choice accuracy is hence represented across all cortical layers as accumulating evidence across the entire trial length[48] independent of the stimulus-dominated auditory response (Fig. 7 and Supplementary Fig. 5).

Altogether, our study demonstrates that the auditory cortex population activity reflects the task complexity at hand, as well as the choice accuracy of the animal. Motor initiation has a stronger impact on cortical activity during detection training, where other

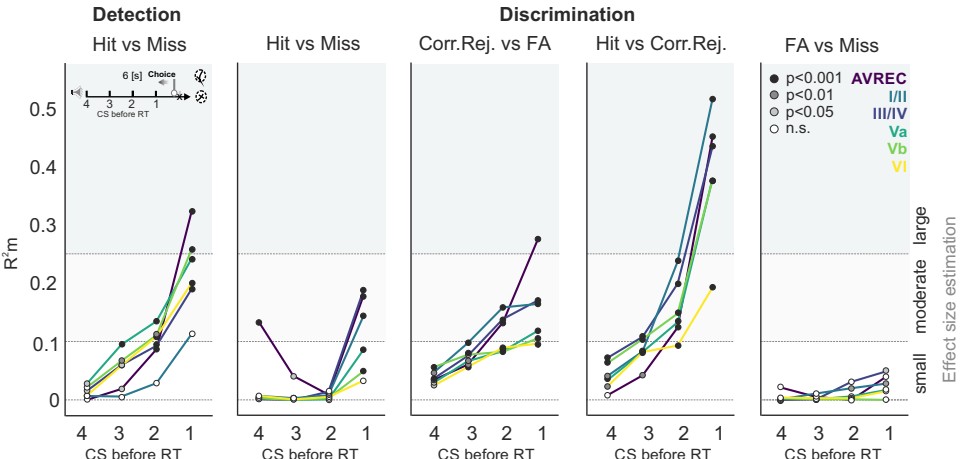

**Fig 7 Time-resolved GLMM-based effect sizes of behavioral outcomes reflecting accumulating evidence over the trial duration.** GLMM-based $R^2$m values for behavioral choices are plotted for time bins before an animal's reaction (small inset top left). Dashed lines indicate small, moderate, and large effect sizes, while the color of circles indicates the corresponding $p$ value of each GLMM (black: $p < 0.001$, dark gray: $p < 0.01$, light gray: $p < 0.05$ and white: n.s.). We found $R^2$m values to generally increase over the trial duration until a behavioral choice option was made. During detection, infragranular layers Va–VI showed moderate $R^2$m values even at 2–3 CS+ presentations before the actual response was commuted. Layers III/IV and I/II only allowed moderate predictions of the animal's choice at the CS+ presentation preceding the reaction. During discrimination, the predictability between hits and misses were considerably less pronounced and time-resolved. Activity in cortical layers I/II and III/IV, however, allowed to correctly predict the occurrence of correct rejections of up to three CS presentations before the animal's reaction. Such temporally dispersed evidence was particularly pronounced in upper layers in contrast to false alarm trials. Largest effects were found when comparing hit vs. correct rejection responses revealing the accumulating evidence of choice accuracy over the trial duration. For incorrect decisions, the model showed no change of low predictability over the trial length.

task-dependent features, such as coding of the contingency, are absent. During the more complex discrimination task, other factors also affect cortical activity. Overall, our results show that the layer-specific population activity in sensory cortex is highly dependent on the task design and reflects the performance of the individual subject in agreement with an accumulating sensory evidence in auditory cortex ultimately leading to a behavioral choice[8,48].

In conclusion, previous work and the current study show that neuronal activity in the primary auditory cortex encodes sounds in ways that are directly relevant to behavior. We found that the entire ensemble activity of the A1 columnar circuits closely represented task-relevant stimulus features, the task rule, and behavioral choice variables suggesting its instructive role for auditory-guided decision-making. While infragranular layers dominated the cortical processing modes during action selection within a detection context, supragranular layers gained relevance when stimulus features needed to be integrated during discrimination. Our study thereby expands our understanding of the layer-specific cortical circuit processing modes which code task-relevant information in order to guide sensory-based decision-making and behavioral adaptation during strategy change. We have now begun to reveal the functional computations performed by single neurons and of the local and long-range cortical networks within which they are integrated[49]. Future studies will enunciate the more widespread brain networks for mediating perceptual decision-making, in which the A1 circuitry reflects only one important hub.

## Methods

**Statement of compliance with ethical regulations.** Experiments were carried out with adult male Mongolian gerbils (*Meriones unguiculatus*, 4–8 months of age, 70–90 g body weight, total $n = 9$). All experiments presented in this study were conducted in accordance with ethical animal research standards defined by the German Law and approved by an ethics committee of the State of Saxony-Anhalt.

**Surgery and chronic implantation under electrophysiological control.** For chronical in vivo electrophysiological recordings a multichannel electrode

(NeuroNexus, A1x32-6 mm-50-177_H32_21mm; linear array of 32 channels with 50 μm distance each) was surgically implanted into the A1. Gerbils were initially anesthetized by an intraperitoneal (i.p.) injection (0.004 ml/g) consisting of 45% ketamine (50 mg/ml, Ratiopharm GmbH), 5% xylazine (Rompun 2%, Bayer Vital GmbH), and 50% of isotonic sodium-chloride solution (154 mmol/1, B. Braun AG). Anesthesia during the surgery was maintained with around 0.15 ml/g h ketamine i.p. infusion. Anesthetic status was regularly checked (10–15 min) by the paw withdrawal-reflex and breathing frequency. Body temperature was continuously measured and kept stable at 34 °C. The primary field A1 of the right auditory cortex was exposed by a small trepanation through the temporal bone (Ø 1 mm). This avoids tissue damage and guarantees stable fixation of the implanted electrode on the skull. Another small hole for an initial reference wire (stainless steel, Ø 200–230 μm) was drilled into the parietal bone on the contralateral side. Animals were head-fixed with a screw-nut glued to the rostral part of the exposed nasal bone plate by UV-curing glue (Plurabond ONE-SE and Plurafill flow, Pluradent) that was temporally attached to a metal bar. The recording electrode with a flexible bundle between shaft and connector was inserted perpendicular to the cortical surface into A1 via the small hole.

During the implantation animals were placed in a Faraday-shielded and acoustic soundproofed chamber. Sounds were presented from a loudspeaker (Tannoy arena satellite KI-8710-32) at 1 m distance to the animal. For verification of the implantation site in A1, a series of pure tones covering a range of at least 7 octaves were presented (0.25–32 kHz; tone duration 200 ms, interstimulus interval (ISI) 800 ms, 50 pseudorandomized repetitions, sound level 65 dB SPL. Stimuli were generated in Matlab (MathWorks, R2006b), converted into an analog signal by a data acquisition card (sampling frequency 1 kHz, NI PCI-BNC2110, National Instruments), rooted through an attenuator (gPAH Guger, Technologies), and amplified (Thomas Tech Amp75). A measurement microphone and conditioning amplifier were used to calibrate acoustic stimuli (G.R.A.S. 26AM and B&K Nexus 2690-A, Bruel&Kjaer, Germany).

Tone-evoked LFPs were recorded with the multichannel array, preamplified 500-fold and band-pass filtered (0.7–300 Hz) with a PBX2 preamplifier (Plexon Inc.). Data were then digitized at a sampling frequency of 1 kHz with the Multichannel Acquisition Processor (Plexon Inc.). Recordings of tone-evoked responses were taken around 30 min after implantation and before the final fixation of the electrode. After 30 min of laminar recordings, to allow for signal stabilization and verification of the tonotopic location, the electrode and connector (H32-omnetics) were glued onto the animal's skull with UV-glue. Before enclosing the exposed A1 with UV-glue an antiseptic lubricant (KY-Jelly, Reckitt Benckiser-UK) was applied to the exposed cortex. After the surgery, the wounds were treated with the local antiseptic tyrothricin powder (Tyrosur, Engelhard Arzneimittel GmbH & Co.KG). Directly after the surgery and over the next 2 days, animals received analgesic treatment with Metacam (i.p. 2 mg/kg bw; Boehringer Ingelheim GmbH) substituted by 5% glucose solution (0.2 ml). Animals were allowed to recover for at least 3 days before the first session of awake electrophysiological recording.

**Characterization of the recording location in A1 during awake—passive listening.** After the recovery period, animals were placed in a one-compartment box in an electrically shielded and sound-proof chamber in order to recharacterize the tuning properties of the chronically implanted electrode. Acoustic stimuli were presented in a pseudorandomized order of pure tone frequencies covering a range of 7 octaves (0.25–16 kHz; tone duration: 200 ms, ISI 800 ms, 50 pseudorandomized repetitions, sound level 70 dB SPL), while laminar LFP signals were recorded. Based on the measurements, we observed a rather flat frequency tuning of the dominant early synaptic inputs in the passively listening gerbil before and after each training phase[50] (detection and discrimination) indicating that tonotopic plastic changes are not the main source of our findings (Supplementary Fig. 3).

**Shuttle-box training and behavioral paradigm.** Behavioral paradigm: Operant conditioning was trained in a two-way avoidance shuttle-box task (see Fig. 1). The shuttle-box (E15, Coulbourn Instruments) was placed in an acoustically and electrically shielded chamber and contained two compartments separated by a hurdle (3 cm height). We trained animals ($n = 9$) twice a day with a break of at least 5 h in between both training sessions. In each session subjects were allowed to habituate for 3 min within the shuttle-box. In the first training phase two pure tones with frequencies 1 and 4 kHz were presented both as "Go" conditioned stimuli (CS+). Subjects needed to detect any tone event and respond with a compartment change in order to avoid a mild foot individualized shock (200–500 µA) presented as the unconditioned stimulus (US). We, therefore, call this phase the detection phase. Within each trial (12–15 s), the CS+ tones were repeatedly presented (tone duration 200 ms, ISI of 1.5 s, 70 dB SPL) in a 6-s observation window during which subjects are required to change the compartment in order to make a correct "hit" response. When subjects shuttled into the other compartment in response to the CS before US onset, this was counted as CR. In case animals did not show a CR within the 6-s observation window this defined a so-called miss trial. Here, the animal received an overlapping presentation of the CS+ and the US until an escape to the other compartment terminated the US/CS presentation. Subjects thereby learned to escape the aversive foot shock within a couple of trials[51]. In each session we presented each CS+ for 30 times in a pseudorandomized order. The time point at which we changed the task rule from detection to discrimination was oriented at the behavioral performance of each subject individually. For the detection task, which consists only of "Go" trials, the overall task performance of each animal was derived by the $d'$ values for each session based on signal detection theory. We calculated the $d'$ as the differences of the z-transforms of the hit rate and the z-transform of the relative intertrial shuttles (ITS) derived from the inverses of a standardized normal distribution function[51]:

$$d' = Z(hits) - Z(ITS), \qquad (1)$$

Once animals reached a stable performance of $d' > 1$ (criterion threshold) for three consecutive training sessions, we introduced a change of the task rule and switched to a discrimination task by assigning the former 4 kHz "Go" tone with a "NoGo" (CS−) contingency ($n = 8$; one subject excluded due to epileptic seizure during training). Subjects needed to report on the "NoGo" condition by staying within the compartment to avoid an US, which we call a "Corr. Rej." In "NoGo" trials, animals had to stay in the compartment for 12–15 s, while the CS− was continuously played with an ISI of 1.5 s to prevent animals from developing a time estimate of the observation window length over the long training period. If subjects incorrectly crossed within this 12–15 s, the behavioral choice was counted as "FA."

Conditioned stimuli (CS): As CS, we used auditory pure tone stimuli, which were generated in Matlab (MathWorks, R2012b), converted into an analog signal by a data acquisition card (NI PCI-6733, National Instruments), rooted through an attenuator (gPAH Guger, Technologies), and amplified (Black Cube Linear, Lehman). Two electrostatic loud speakers were positioned 5 cm at both sides of the shuttle-box. A measurement microphone and conditioning amplifier were used to calibrate acoustic stimuli (G.R.A.S. 26AM and B&K Nexus 2690-A, Bruel&Kjaer, Germany).

Unconditioned stimuli (US): The mild foot shock (US) was conditionally delivered by a grid floor and generated by a stimulus generator (STG-2008, Multichannel Systems MCS GmbH). Depending on the individual animal sensitivity and performance the shock intensity was adjusted (starting at 200 µA) in steps of 50 µA until the escape latencies were below 2 s, in order to achieve a successful association of CS and US[51].

Data analysis: We recorded all compartment changes during the habituation phase and the training phase. Reaction times of CR's, escape latencies and the number of ITS were recorded. The choice outcomes were characterized as hit, miss, FA, and Corr. Rej. depending on the animals' behavior and the contingency of the stimulus (see Fig. 1a, right). To evaluate the training progress, we calculate the averaged CR rates as a function of sessions (Fig. 1b) and $d'$ learning curves (Supplementary Fig. 1).

**Multichannel recordings during training.** Multichannel recordings were performed with connecting the head-connector of the animal to a preamplifier (20-fold gain, band-pass filtered, HST/32V-G20; Plexon Inc.) and a data acquisition system (Neural Data Acquisition System Recorder Recorder/64; Plexon Inc.). The cable harness was wrapped by a metal mesh for bite protection. Tension of the cable was relieved by a spring and a turnable, motorized commutator (Plexon Inc.)

that permits free movement and rotation of the animal in the box. Broadband signals were recorded continuously using a preamplifier (Plexon REC/64 Amplifier; 1Hz-6 kHz) during the training with a sampling frequency of 12 kHz. LFPs were sampled with 2 kHz, visualized online (NeuroExplorer, Plexon Inc. Recording Controller) and stored offline for further analysis. To avoid ground loops between recording system, shuttle-box and the animal we ensured proper grounding of the animal via its common ground and left the grid floor on floating voltage.

**Analysis of electrophysiological data.** Current source density (CSD) analysis: Based on the recorded laminar LFPs, the second spatial derivative was calculated yielding an estimate of the CSD distribution, as seen in equation:

$$-CSD \approx \frac{\delta^2 \Phi(z)}{\delta z^2} = \frac{\Phi(z + n\Delta z) - 2\Phi(z) + \Phi(z - n\Delta z)}{(n\Delta z)^2}, \qquad (2)$$

where $\Phi$ is the field potential, $z$ is the spatial coordinate perpendicular to the cortical laminae, $\Delta z$ is the spatial sampling interval, and $n$ is the differential grid[52]. LFP profiles were smoothed with a weighted average (Hamming window) of nine channels which corresponds to a spatial kernel filter of 400 µm[21].

CSD distributions reflect the local spatiotemporal current flow of positive ions from extracellular to intracellular space evoked by synaptic populations in laminar neuronal structures. CSD activity thereby reveals the spatiotemporal sequence of neural activation across cortical layers as ensembles of synaptic population activity[21,53]. One advantage of the CSD transformation is that it is reference-free and hence less affected by far-field potentials and referencing artifacts. It allows to observe the local synaptic current flow with high spatial and temporal precision[54]. Current sinks thereby correspond to the activity of excitatory synaptic populations, while current sources mainly reflect balancing return currents. The CSD thus provides a functional readout of the cortical microcircuitry function, encompassing a wider, mesoscopic field of view than for instance single- or multiunit approaches[55]. Early current sinks in the auditory cortex are, therefore, indicative of thalamic input in granular layers III/IV and infragranular layers Vb/VI[21,56–59]. In order to describe the overall columnar processing, the CSD profiles were transformed by averaging the rectified waveforms of each channel:

$$AVREC(t) = \frac{\sum_{i=1}^{n} |CSD_i|(t)}{n}, \qquad (3)$$

where $n$ is the number of recording channels and $t$ is time. The AVREC reflects the temporal overall local current flow of the columnar activity[60,61].

Data preprocessing: Single-trial data were analyzed via a custom-written graphical user interface (MathWorks, R2016a & R2017b) that visualized the LFP, CSD, and behavioral parameters to inspect and mark two types of artifacts: (1) affected recording channels and (2) foot shock or movement induced signal clipping and distortions. Affected channels were substituted by a linear interpolation method across neighboring, unaffected channels on the level of the LFP[21]. Shock induced clipping was rejected from the overall signals. Trials with artifacts due to extreme movements were also discarded from further analysis.

Extraction of signal parameters: Cortical layers were assigned to the recording channels based on the averaged auditory-evoked spatiotemporal CSD current flow in response to the first CS presented during a session and compared to the awake measurement location (Supplementary Fig. 2). Early dominant current sinks in the auditory cortex are indicative of thalamic inputs in granular layers III/IV and infragranular layers Vb/VI[21,56] and allow to identify supragranular layers I/II and infragranular layers Va and VI in the CSD recordings (Fig. 1d). We determined trial-by-trial RMS values of averaged CSD traces within each of the five cortical depths from tone onset of each CS presentation in a time window of 500 ms. Also, we calculated the RMS value of the AVREC within the same time windows for the corresponding overall columnar response. We did not inspect the time-points after a CR, as the CS presentation was terminated. For statistical analysis, single-trial values were z-norm. across trials.

**Statistics and reproducibility.** Statistical test of variance: Statistical difference between groups was tested by one-factorial rmANOVA to account for the hierarchical structure of the data using R Studio (R 3.5.1.). Replication was tested indirectly by using large sample sizes over repeated measures and using appropriate statistical methods that explicitly control for within and across sample size variability. We used an overall significance level of $\alpha = 0.05$ and paired-sample $t$-tests with a Holm-adjusted significance level[62] for posthoc testing. Before testing, data was generally z-norm. within each animal and session. The generalized eta squared $\eta^2_{gen}$ is reported as measure of effect size calculated using the R package DescTools[63,64]. In general, we interpret effect sizes to be small for $\eta^2_{gen} \leq 0.1$, as medium for $0.1 < \eta^2_{gen} < 0.25$, and large for $\eta^2_{gen} \geq 0.25$[64].

Mixed-effects logistic regression: For statistical comparison between two-choice classes, parameters of interest were analyzed on a single-trial level using GLMM with a logistic link function[23]. GLMM calculation in R Studio (R 3.5.1) was done with the lme4 package for model estimation and ggplot2 and sjplot for plotting.

Logistic regression was used for predicting the probability of the binary (0/1) dependent variables $\pi_i = E(y_i)$. The predictions were then wrapped by the logistic

link function:

$$g(x) = \frac{1}{1 + \exp(-x)} \tag{4}$$

To map the predictions of the model to the interval between 0 and 1. In the mixed-effects logistic regression, random effects were additionally introduced to model subject-specific variance by:

$$g(E(\boldsymbol{y}_i)) = \boldsymbol{X}_i\boldsymbol{\beta} + \boldsymbol{Z}_i\boldsymbol{v}_i, \tag{5}$$

where $\boldsymbol{y}_i$ is the vector of all responses of the $i^{th}$ animal, $\boldsymbol{X}_i$ and $\boldsymbol{Z}_i$ are design matrices, $\boldsymbol{\beta}$ the fixed effects and $\boldsymbol{v}_i$ the animal-specific random effects. The parameters of the estimated model can be interpreted as logarithmic odds ratios $log(\pi_{ij}/1 - \pi_{ij})$, where $\pi_{ij}$ corresponds to the probability of the outcome to be 1 for animal $i$ in trial $j$. The GLMM thus allows for an intuitive interpretation of its predicted values (choice probabilities) and its estimated coefficients (logarithmic odds ratios). As such, GLMMs are optimally suited to compare data on a trial-by-trial level while accounting for within-subject variability. Random intercepts were introduced to account for the general variability in overall activity across subjects and random slopes to allow for the fixed effect to vary between animals. We z-norm. the AVREC RMS values for the GLMM to facilitate the estimation procedure.

Evaluation of the model: Calculation of the marginal ($R^2$m) and conditional ($R^2$c) coefficient of determination was done using the MuMIn package[65]. The $R^2$m represents the variance in the dependent behavioral variable (on the logistic link scale) explained by the fixed effect of the respective CSD variable (across subjects), while the $R^2$c reflects the total variance explained by the model's fixed and random effects, respectively[66]. In a binary GLMM, the $R^2$m is independent of sample size and dimensionless, which allows comparing fits across different datasets[66]. An $R^2$m of 0.2 thus means that 20% of the variance in the binary outcome can be explained by the cortical activity variable, which was used as the model's predictor. If the corresponding $R^2$c is 0.35, the whole model explains 35% of the variance, meaning that an additional 15% of the variance in the outcome can be explained by the variability between animals. The $R^2$m can hence be used to estimate the effect size, which we did in accordance with the $\eta^2_{gen}$ from rmANOVA tests and report small effects for $R^2$m ≤ 0.1, medium effects for $0.1 < R^2$m $< 0.25$, and large effects for $R^2$m ≥ 0.25[64].

**Reporting summary**. Further information on research design is available in the Nature Research Reporting Summary linked to this article.

## Data availability

Preprocessed data are available via the following GitHub repository: https://github.com/mzempelt/Zempeltzi_etal_2020. Raw data (plx-files and converted Matlab-files) are available upon request from the corresponding authors.

## Code availability

Documented analysis code for the entire data analysis presented (R studio) is available via the following GitHub repository: https://github.com/mzempelt/Zempeltzi_etal_2020.

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

## Acknowledgements

We would like to thank Kathrin Ohl and Anja Guerke for their technical assistance. This project was founded by the Deutsche Forschungsgemeinschaft (DFG SFB 779) and by the Leibniz Association (LIN Postdoctoral Network, LPN).

## Author contributions

F.W.O., M.D., and M.F.K.H. designed research and conceived the study. M.M.Z., M.G.K.B., and S.A. collected experimental data. M.M.Z., M.K., C.G., K.E.D., S.M., and L.S. analyzed data. M.D. and M.F.K.H. supervised experiments and data analysis. M.D. established the setup and stimulus protocols. M.M.Z. and M.F.K.H. wrote the manuscript. M.M.Z., C.G., M.G.K.B., K.E.D., F.W.O., M.D., and M.F.K.H. discussed and all authors reviewed the manuscript

## Competing interests

The authors declare no competing interests.
