## [Peer Review File · Communications Biology]

Reviewers' comments:

Reviewer #1 (Remarks to the Author):

The paper by Zempeltzi et al. reports layer-specific dynamics in mesoscopic auditory cortical activity, which correlate with task rule and decision outcome in a behavioural paradigm. Based on CSD analyses, the authors describe differences in the laminar activation patterns between circumstances in which animals are trained to "detect" or "discriminate" auditory stimuli. The overall cortical activation differs in the "detect" or "discriminate" scenarios in an interesting layer-specific manner, based on an adroitly constructed GLMM. The manuscript is well-written and the authors manage to convey their ideas clearly. Overall, I find the results compelling, and I think the data advance our understanding of how behaviour based on audition might rely on the neural circuitry within primary sensory cortices. I still have a few comments that could help the authors improve the quality and transparency of the paper.

Introduction

What do the authors mean by their sentence (1st paragraph): "Some studies have suggested..."? In view of the literature they use to substantiate what is said here (refs. 13-15), it is a bit unclear to me what the differences are and what type of information is related to these differences in the vertical axis. Could you be more specific in this part of the introduction?

On a second note, I suggest removing "In agreement" at the beginning of the next sentence, maybe in favour of "In addition". From attention and task-related modulations in human A1 (measured with non-invasive techniques), it does not necessarily follow that there exist laminar differences (even if there are).

The rationale of the work is stated in the first paragraph of the Introduction. However, it could be helpful to mention explicitly the main objective(s) of the study, perhaps at the height of the second paragraph, and before describing the methodology used to achieve it (and the outcomes). I furthermore have the feeling that the segment (2nd paragraph) "Based on single-trial [...] during inadequate choices" is a rather long description of the results, which sounds like it belongs more in that section, the Abstract, or even the Discussion. Perhaps the segment "Our study revealed..." is an already good wrapping of the study for the Introduction, allowing the readers to find the details further in the text. This is ultimately the authors' decision.

Methods:

The fact that there's a flat frequency tuning is a bit surprising to me, given the properties of primary auditory cortices in mammals (also, neuronal activity in gerbil A1 shows clear tuning, see e.g. (Schaefer et al., 2015; Schaefer et al., 2017)). Why is the case that the tuning is poor here? Is this due to the integrative nature of LFP/CSD signals?

"Extraction of signal parameters": I think channels are rather assigned (post-hoc) to the cortical layers, maybe not the other way around. I would also suggest considering "associated" as a term. For the ANOVA and the t-tests, how was the normality of the data confirmed? Is it possible to venture a qualitative interpretation of the effect size?

Results:

The way the data are depicted in some figures (e.g. Fig. 2B) could profit from showing the sample size in the panels. Occasionally, in boxplots for example, it is not directly intuitive whether we're dealing with data from the 9 animals (i.e. one sample per animal), or from all trials. Additionally, in Fig. 1B, what are the error bars showing?

First section, third-to-last sentence: Wouldn't it always be the case that reaction times come after the first CS presentation? In the figure legend for this part (Fig. 1C), it is mentioned that most CR timings are after the second CS. This is less trivial. Also, from the figure, it is not evident to me that in the

case of "discrimination", reaction times were equally distributed during the last 4.5 s. Rather, it appears that the CR count (for hits) starts waning past the penultimate CS presentation.

"Task rule impacts on the..." subsection: I appreciate the insets in Fig. 2B, summarizing the statistical results obtained. I imagine that the authors have access to the AVREC traces whose RMS values are summarized in the boxplots. I encourage editing the insets so that instead of a schematic, they would show the actual AVREC over time (even a summary of it; e.g. a grand average of the traces across trials). This way the reader would have access to the data used in the analysis at least visually. (also applicable for Fig. 3B)

In Figure 3: Why is correct rejection the lowest? Could you speculate about this?

Subsection "Representation of contingency...": Is there any qualitative way of interpreting the R2m as an effect size?

Second paragraph: 500 ms window around or starting at the CS presentation? What was the criterion for including the CS if the animal reacted within that 500 ms window? How could this have influenced the results of the GLMM here if the animal switched shuttles in this period?

Discussion:

Subsection "Supragranular layer activity better...", second paragraph (also related to data in Fig. 4A,B, right): Can the authors discuss a bit further these results? Why only input layers for discriminating the tone frequency? Is this because of gain modulation due to behavioural relevance? Could plasticity effects (e.g. centripetal shifts in the overall frequency tuning in cortex around the 1 kHz CS+ (see Sakai and Suga, 2002; ref 39) explain these results, particularly in the "discrimination" phase?

Same paragraph, final statements (also related to Fig. 5A, centre): I would be interested in reading the authors' take on top layer activity predicting for hit/miss, and their beliefs on the possible circuitry beyond crosscolumnar communication in A1. Could top layers relay information to other "higher" cortical areas (e.g. prefrontal cortex) so that a decision is made? Or is it the case that they're receiving feedback? What can be speculated in terms of the existing literature?

Subsection "Choice accuracy...": the data regarding the 500-1000 ms period after tone presentation sounds exciting. Why not show it a supplement?

Minor comments:

Some sentences read a bit long occasionally, and therefore give the impression of missing commas that could improve intelligibility.

Introduction:

Last sentence: what do you mean by "instructive mediator"?

Methods:

To simplify the task of a reader not familiar with NeuroNexus nomenclature, please mention the number of channels of the probe and the spacing between them.

Why was the frequency range different between A1 frequency tuning characterization in anaesthetized and awake animals (i.e. 0.25-32 and 0.25-16 kHz)?

Sentence before eq. 1: "...inter-trial (ITS).." is there a word missing? Shuttles? (the acronym seems to be redefined further ahead in "Data analysis").

Subsection "Acoustic stimuli": sentence "Two electrostatic..." seems incomplete.

Sometimes the tense of some verbs is present. Consider changing those to past tense. (e.g. first sentence, "Data analysis" subsection).

"Multichannel recordings during training", last sentence: "leave" for "left"?

CSD analysis section, second paragraph, third sentence? "is" missing?

"Mixed-effects logistic regression": Missing period mark before the second appearance of GLMM. (right after ref. 33).

Results:

Fig. 2B: insets are a little bit unaligned with each other.

"Auditory cortex represents...", third sentence: change "report" for "reported".

Sentence starting with "Holm-corrected...": should "value" here be plural?

Subsection "Representation of contingency...", 3rd paragraph: change "was also predicting" for "predicted".

Discussion:

Subsection "Correlates of motor...", 2nd sentence: remove the comma after "hypothesize".

Reference list:

Schaefer, M.K., Hechavarria, J.C., and Kossl, M. (2015). Quantification of mid and late evoked sinks in laminar current source density profiles of columns in the primary auditory cortex. *Front Neural Circuits* 9, 52.

Schaefer, M.K., Kossl, M., and Hechavarria, J.C. (2017). Laminar differences in response to simple and spectro-temporally complex sounds in the primary auditory cortex of ketamine-anesthetized gerbils. *PLoS One* 12, e0182514.

Reviewer #2 (Remarks to the Author):

The study by Zempeltzi et al., uses laminar CSD in auditory cortex of freely moving gerbils to understand if and how task-related variables are encoded across layers. The study is extremely carefully executed, and uses an elegant generalized linear-mixed effects model to interpret the results. The results are important for the field by carefully showing the enriched encoding of choice in supergranular layers during discrimination.

I have no major concerns.

Minor concerns:

D' learning curves for the data in Figure 1B, right, would be valuable information.

Much of the analyzes focus on comparing e.g. physiological parameters on hits to CRs, or FAs to misses. It would be helpful to do full 2-way ANOVAs on all four conditions.

The y axes in Figure 2B are not sufficiently labeled with tick marks

Labels in Figure 5B and 6B of what each of the 3 'cylinders' correspondes to would be helpful.

The gray traces in 5A and 6A are too light to see. They're line weight should be increased.

More justification should be presented to support that they are recording truly in A1 (as opposed to AAF or a belt region).

Reviewer #3 (Remarks to the Author):

This study elegantly show using CSD analysis in awake behaving mice performing a detection or a discrimination task in a shuttle box, that task-related decisions strongly participate in the recruitment of cortical activity, and that in some way it is possible to predict the decision of the animal based on CSD responses.

The data is sound and valuable, and although we know already that task parameters such as reward or engagement are encoded in auditory cortex, the fact there is some information about the decisions

of the animal is new. That said, we do not know here whether this information is contained in the spiking activity of auditory cortex neurons, as CSD is mainly reflecting synaptic activity, including extrinsic inputs, such as feedback.

Despite the clarity of the study and of the manuscript, the analysis and presentation of the results is very compact and could be extended. Specifically:

- Prediction of decisions (or sound identity) is only based on GLMs applied to the 500ms before an escape or an absence of escape decision, showing that decision is a predicting factor of the CSD. It would be very informative to train classifiers and measure what is the fraction of decision that can be correctly predicted.

- The GLM measure hides a little the differences between layers that the author often mention. From CSD examples, it seems that all layers are up modulated by Go decisions, with small differences in the degree of modulation. To better appreciate the level of cross-layer difference, the authors could plot the difference between data and a homogenous modulation model.

- Very little is said about timing. How far back can you predict a decision. Maybe using classifiers, the authors should provide a curve quantifying the progression of the prediction over time until and eventually after the decision of the animal.

Response Letter for COMMSBIO-19-1888

Manuscript: Task rule and choice are reflected by layer-specific processing in rodent auditory cortical microcircuits

Please find below the point-by-point responses (in blue) to each comment of the reviewers (in black).

Response to Reviewer 1

The paper by Zempeltzi et al. reports layer-specific dynamics in mesoscopic auditory cortical activity, which correlate with task rule and decision outcome in a behavioral paradigm. Based on CSD analyses, the authors describe differences in the laminar activation patterns between circumstances in which animals are trained to “detect” or “discriminate” auditory stimuli. The overall cortical activation differs in the “detect” or “discriminate” scenarios in an interesting layer-specific manner, based on an adroitly constructed GLMM. The manuscript is well-written and the authors manage to convey their ideas clearly. Overall, I find the results compelling, and I think the data advance our understanding of how behavior based on audition might rely on the neural circuitry within primary sensory cortices. I still have a few comments that could help the authors improve the quality and transparency of the paper.

Reply to Reviewer #1:

We would like to thank the reviewer for the careful reading and the appreciation of our study. Please find changes according to the fruitful comments and suggestions raised by the reviewer in our revised manuscript. We are convinced that the updated version of the manuscript significantly increased in clarity and rationale of our applied procedures.

Introduction

What do the authors mean by their sentence (1st paragraph): “Some studies have suggested...”? In view of the literature they use to substantiate what is said here (refs. 13-15), it is a bit unclear to me what the differences are and what type of information is related to these differences in the vertical axis. Could you be more specific in this part of the introduction?

We now more precisely state on page 1, 2nd column:

“Some studies have suggested layer-specific differences in the representation of auditory information along the vertical axis of the auditory cortex with granular layers revealing more accurate tonotopic response properties due to the dominant lemniscal inputs compared to supragranular and infragranular layers¹³⁻¹⁵”

On a second note, I suggest removing “In agreement” at the beginning of the next sentence, maybe in favour of “In addition”. From attention and task-related modulations in human A1 (measured with non-invasive techniques), it does not necessarily follow that there exist laminar differences (even if there are).

We made the according change.

The rationale of the work is stated in the first paragraph of the Introduction. However, it could be helpful to mention explicitly the main objective(s) of the study, perhaps at the height of the second paragraph, and before describing the methodology used to achieve it (and the outcomes). I furthermore have the feeling that the segment (2nd paragraph) “Based on single-trial [...] during inadequate choices” is a rather long description of the results, which sounds like it belongs more in that section, the Abstract, or even the Discussion. Perhaps the segment “Our study revealed...” is an already good wrapping of the study for the Introduction, allowing the readers to find the details further in the text. This is ultimately the authors’ decision.

We have rephrased and streamlined the introduction (and also the abstract) according to the well-received reviewer's comments (and due to a new analysis incorporated in response to the third reviewer). Please find the revised parts indicated in yellow in the new version.

Methods:

The fact that there's a flat frequency tuning is a bit surprising to me, given the properties of primary auditory cortices in mammals (also, neuronal activity in gerbil A1 shows clear tuning, see e.g. (Schaefer et al., 2015; Schaefer et al., 2017)). Why is the case that the tuning is poor here? Is this due to the integrative nature of LFP/CSD signals?

Pure-tone derived tuning widths on the level of LFP/CSD measures differ between awake and anesthetized recordings. We have recently published a study on a preprint server (currently under peer review), where we compared CSD-based frequency tunings in the awake A1 and under ketamine anesthesia from animals of the same data set (Deane et al, 2019). Please find below the peak amplitude tuning curves of the dominant layer III/IV sink measured under ketamine anesthesia (during implantation of the recording electrodes) and the awake condition (Figure R1, *left*). The comparison reveals the stronger frequency tuning under ketamine anesthesia compared to awake recordings.

In order to further reveal the implantation of the electrode within the primary auditory cortex, mean onset latencies are a commonly used indicator reflecting early onset activity relayed via the subcortical auditory pathway. We did not observe significant differences in the onset latency of the granular sink in layer III/IV between both conditions (Figure R1, *right*). For stimulation with the best frequency, onset latencies were in the range of around 18ms. This is typically reported for CSD-derived onset latencies obtained from recordings in the field AI in primary auditory cortex of the Mongolian gerbil (cf. Happel et al., 2010). We also now reference the study by Deane et al. (2019) in the revised version and explain the influence of the anesthesia status on the CSD-derived frequency tuning in A1 within the caption of the corresponding Supplemental Figure 4.

Figure R1. Tuning curves of the mean peak amplitude and the mean onset latencies of the dominant sink in thalamocortical recipient layers III/IV between the awake and anesthetized group. *Left*, Mean peak amplitude tuning curves (centered to the best frequency; BF) showed stronger tuning during ketamine anesthesia compared to awake recordings due to a rampant recurrent excitation (cf. Deane et al., 2019). This is reflected by significant main effects of a two-way rmANOVA for group (G) and frequency (F), as well as a significant interaction (X) of both factors (indicated in the figure panel). *Right*, We did not observe significant differences of the earliest onset latency of the dominant layer III/IV sink between both groups. Stable onset latencies for BF stimulation in the range of ~18ms in both conditions matches the parameters reported before for the field A1 of primary auditory cortex in the Mongolian gerbil (cf. Happel et al., 2010, Happel and Ohl, 2017). Correspondingly, an rmANOVA only reported significant effects for the factor frequency, but not for group or their interaction. Figures adapted from Deane et al., (2019; currently in revision).

Correspondingly, we now state on page 3, 1st column, 2nd par.:

“Based on the measurements, we observed a rather flat frequency tuning (Supplementary Fig. 2b) of the dominant early synaptic inputs in the passively listening gerbil before and after each training phase (detection and discrimination) indicating that tonotopic plastic changes are not the main source of our findings (Supplementary Fig. 4).”

“Extraction of signal parameters”: I think channels are rather assigned (post-hoc) to the cortical layers, maybe not the other way around. I would also suggest considering “associated” as a term. For the ANOVA and the t-tests, how was the normality of the data confirmed? Is it possible to venture a qualitative interpretation of the effect size?

The reviewer asked for the normality assumption of the data. We now checked the residuals of each statistical test visually using Q-Q plots. In very most tests of our study, those showed no systematic bias suggesting a normal distribution of the residuals (for example, see Figure R2-left below as a representative example; plotted parameter: AVREC RMS value for 3rd stimulus in the discrimination phase). A few individual tests showed some deviations, as shown in Figure R2-right (example parameter shown here AVREC RMS value for 4th stimulus in the discrimination phase). We opted for a repeated-measures ANOVA in the initial manuscript, as it is a common and accepted test for data with multiple groups and repeated testing. Nevertheless, we wanted to control if the ANOVA results are robust. We compared each individual statistical test of the study with a nonparametric test, the Wilcoxon signed-rank test and Friedman tests with Holm-corrected Wilcoxon tests for post-hoc comparison. Our initial rmANOVA results showed the same statistical significances across all applied tests, except of one single post-hoc comparison between false alarm and correct rejections in the discrimination phase (for AVREC RMS value of the 1st stimulus; not relevant for our interpretation of the study). This reveals the robustness of our statistical test results, and given the minor difference, which is not influencing the interpretation of the data, we would like to suggest keeping the ANOVA-based statistical analysis in our paper.

Figure R2. Representative examples of Q-Q-plots of the residuals revealing normality in our data set to a large degree. For the very most model residuals, we found the sample and theoretical residuals to be aligned suggesting a normal distribution of the data (left). In some cases, smaller deviations were observed, as in the example shown on the right.

The other point of the reviewer was a qualitative interpretation of the effect size of our results, which is applicable for the rmANOVA tests, as well as the GLMM results. Accordingly, Bakeman (2005) proposed η_{gen}^2 (generalized eta squared) as the preferred effect size for analyses of variance with repeated measures as it is comparable across both between and within-subject designs. It reflects the ratio of the effect variance to the total variance and hence can be interpreted as the proportion of variance in the dependent variable (in our case the electrophysiological parameter: the AVREC RMS value) explained by the independent variable (e.g. tone frequency or the behavioral outcome). Bakeman (2005) proposes to interpret a η_{gen}^2 of 0.02-0.13 as a small effect size, of 0.13-0.26 as medium and above 0.26 as large. We now explicitly state this in the paper and chose boundaries more comprehensible for the reader with small effect sizes for $\eta_{gen}^2 \leq 0.1$, medium effect size for $0.1 < \eta_{gen}^2 < 0.25$ and large effects, when $\eta_{gen}^2 \geq 0.25$. We rephrased the corresponding section in the methods on page 5, 1st column, 2nd par., which now reads:

“The generalized eta squared η_{gen}^2 is reported as measure of effect size calculated using the R package DescTools^{31,32}. In general, we interpret effect sizes to be small for $\eta_{gen}^2 \leq 0.1$, as medium for $0.1 < \eta_{gen}^2 < 0.25$, and large for $\eta_{gen}^2 \geq 0.25$.”

Results:

The way the data are depicted in some figures (e.g. Fig. 2B) could profit from showing the sample size in the panels. Occasionally, in boxplots for example, it is not directly intuitive whether we’re dealing with data from the 9 animals (i.e. one sample per animal), or from all trials. Additionally, in Fig. 1B, what are the error bars showing?

We agree with the reviewer, that it was not directly inferable from the plots, where we showed the full data set and where aggregated data was used. We hence now make this explicit within the captions of each figure. In case of aggregated data, the n=9/8 (detection/discrimination; one animal was not trained in discrimination as the implant fell off meanwhile). GLMM boxplots show unaggregated data and allow us to observe the distribution of the middle 50% of the data (boxes) and the full range (whiskers) including potential outliers (dots). Solely for visualization purposes, we trimmed the range of the boxplots somewhat to the outer end of the longer whiskers. Trial counts per comparison range between 1000 – 3000 trials. Error bars in Figure 1b show the s.e.m. which is now stated in the caption.

First section, third-to-last sentence: Wouldn't it always be the case that reaction times come after the first CS presentation? In the figure legend for this part (Fig. 1C), it is mentioned that most CR timings are after the second CS. This is less trivial. Also, from the figure, it is not evident to me that in the case of "discrimination", reaction times were equally distributed during the last 4.5 s. Rather, it appears that the CR count (for hits) starts waning past the penultimate CS presentation.

We thank the reviewer for their critical reading in this respect. Indeed, we meant that most CRs were happening after the presentation of the second CS in a trial. We corrected this in the revised manuscript. The reviewer is also right with the observation that reaction times during the discrimination phase were most prominently distributed around the 2nd and 3rd CS presentation. This might be due to a training effect over the consecutive training sessions of both phases. Statistical differences between the behavioral outcomes, however, were very comparable when focusing on the 3rd or 4th CS presentation (see Figure 3a of the manuscript). Further, most analyses were based on a reaction-time-based binning. Hence, we therefore believe that the distribution of reaction times does not influence our interpretation of the data.

"Task rule impacts on the..." subsection: I appreciate the insets in Fig. 2B, summarizing the statistical results obtained. I imagine that the authors have access to the AVREC traces whose RMS values are summarized in the boxplots. I encourage editing the insets so that instead of a schematic, they would show the actual AVREC over time (even a summary of it; e.g. a grand average of the traces across trials). This way the reader would have access to the data used in the analysis at least visually (also applicable for Fig. 3B).

We now included the corresponding raw AVREC traces into a new Supplementary Fig. 3, which reflects the illustration from the original paper. We believe that it is more comprehensible to show the illustration in the main Figures 2 and 3 but would like to include the raw data into the manuscript for the reader's inspection.

In Figure 3: Why is correct rejection the lowest? Could you speculate about this?

This is indeed an interesting question, which we can only speculate about. In another recent study, (Nanda et al. 2020) also found lower activity based on LFP measures in the rat parietal cortex during correct rejection trials in an auditory licking Go/NoGo task. They speculate that the higher activity in hit trials may reflect the target identification or the behavioral/motor response activation. They further found differences between miss trials and correct rejections (both choice options without any motor response) and suggest that lower activity during correct rejections might reflect an active response inhibition, while error-response processing might be reflected during miss trials. This might be related also to human findings of a cognitive and motor inhibition during a Go/NoGo task (Smith et al., 2008) We now included those references in the discussion section on page 10, 1st column:

"Several studies also observed an enhanced representation of target stimuli that initiated an auditory-guided motor response in various Go/NoGo discrimination tasks^{10,59,60}. Studies in rats and humans suggested that lower activity during correct rejections might reflect an active inhibition of motor or cognitive responses during a Go/NoGo task^{61,62}. Others, however, found higher cortical recruitment during correct rejections compared to hit trials in a Go/NoGo task in the macaque A1¹¹.

More research is needed to link the potentially inherent neuronal variability to the exact task design at hand, aversive or appetitive reinforcing regimes, and stimulus characteristics which may partially explain contradictory findings^{12,63}.”

Subsection “Representation of contingency...”: Is there any qualitative way of interpreting the R^2m as an effect size?

We believe that this is indeed a strength of our statistical analysis approach, as we can interpret the R^2m given by our GLMM model as an effect size. According to the effect size interpretation for the rmANOVA tests, we have adapted a more comprehensible and corresponding explanation of the interpretation of the R^2m in the methods section on page 5, 2nd column. We now explicitly state ranges of the R^2m for small, medium and large effect sizes following the same rationale as for the η^2_{gen} . Please see corresponding edits on page 5, 2nd column:

“The R^2m represents the variance in the dependent behavioral variable (on the logistic link scale) explained by the fixed effect of the respective CSD variable (across subjects), while the R^2c reflects the total variance explained by the model’s fixed and random effects, respectively³⁸. In a binary GLMM, the R^2m is independent of sample size and dimensionless, which allows comparing fits across different datasets³⁸. An R^2m of 0.2 thus means that 20% of the variance in the binary outcome can be explained by the cortical activity variable, which was used as the model’s predictor. If the corresponding R^2c is 0.35, the whole model explains 35% of the variance, meaning that an additional 15% of the variance in the outcome can be explained by the variability between animals. The R^2m can hence be used to estimate the effect size, which we did in accordance with the η^2_{gen} from rmANOVA tests and report small effects for $R^2m \leq 0.1$, medium effects for $0.1 < R^2m < 0.25$, and large effects for $R^2m \geq 0.25$ ³⁵.”

Second paragraph: 500 ms window around or starting at the CS presentation? What was the criterion for including the CS if the animal reacted within that 500 ms window? How could this have influenced the results of the GLMM here if the animal switched shuttles in this period?

The reviewer raised an important point. In our initial paper, we showed only the dependence of cortical activity and the behavioral outcome in the time window directly preceding the reaction time. In principle, we may have included trials where the conditioned response lies within the 500ms time window after the corresponding stimulus presentation. However, given the distribution of reaction times (see Figure 1c of the manuscript), this was only the case for a minor fraction of trials. We ran the analysis again excluding trials in which animals showed reaction times within the time bin of 500ms and found that the results were the same with the exception of negligible decreases in effect sizes for some analyses.

We further analyzed the other time bins preceding the reaction time in order to reveal the temporal accumulation of such task-related evidence in the cortical activity patterns. Those would not include any trials in which the motor response of the animal directly falls into the window of observation. We have now included this analysis to the revised version of the manuscript, as it has revealed interesting effects on how evidence accumulates during behavioral options (new Figure 7). It further shows that the mere motor activity during a jump does not explain the significant effects revealed by

the GLMM analysis (for example, compare hit vs correct rejection and false alarm vs miss; the two outer right panels)

Discussion:

Subsection “Supragranular layer activity better...”, second paragraph (also related to data in Fig. 4A,B, right): Can the authors discuss a bit further these results? Why only input layers for discriminating the tone frequency? Is this because of gain modulation due to behavioral relevance? Could plasticity effects (e.g centripetal shifts in the overall frequency tuning in cortex around the 1 kHz CS+ (see Sakai and Suga, 2002; ref 39) explain these results, particularly in the “discrimination” phase?

The reviewer raises an interesting point here. During detection, subjects are not required to discriminate between the two conditioned stimuli, although they are behaviorally relevant. So, task engagement alone may not explain the findings. However, in case of discrimination, indeed a differential gain modulation of cortical inputs particularly in dependence on the contingency (rather than frequency) might explain the differences found for the granular input layers. We also compared the frequency tuning curves of the dominant early synaptic input in thalamocortical input layers III/IV in awake, but passively listening animals before and after the detection and discrimination training phases (see Supplementary Figure 4). We found that also after the two training phases, tuning curves in awake and passively listening animals were comparably flat as in naïve and awake animals. Hence, we could not demonstrate considerable plastic shifts of the frequency tuning in thalamocortical input layers. In summary, we believe that the different activation of the auditory cortex in response to the same two pure tone frequencies is explained best by a differential involvement of the auditory cortex in the processing of information about the contingency, sensorimotor planning, and decision making.

We now explicitly state on page 8, 2nd column, end 2nd par:

“Of importance, plastic shifts of auditory tuning properties would not explain our findings (Supplementary Fig. 4), strongly arguing for a response modulation in the auditory cortex by the task-relevant contingency, sensorimotor planning, and decision making.”

Same paragraph, final statements (also related to Fig. 5A, centre): I would be interested in reading the authors’ take on top layer activity predicting for hit/miss, and their beliefs on the possible circuitry beyond crosscolumnar communication in A1. Could top layers relay information to other “higher” cortical areas (e.g. prefrontal cortex) so that a decision is made? Or is it the case that they’re receiving feedback? What can be speculated in terms of the existing literature? Subsection “Choice accuracy...”: the data regarding the 500-1000 ms period after tone presentation sounds exciting. Why not show it a supplement?

Indeed we believe that the crosscolumnar communication beyond A1 integrates diverse information into the processing in sensory cortex, for instance via associative and prefrontal cortex. Given the literature, this includes most likely feedforward and feedback information, as cortical communication is highly recurrent (eg. Fritz et al. 2010; Petrides and Pandya 1988; Sheikhattar et al. 2018). Although anatomical projections towards the prefrontal cortex are strongest from non-primary auditory areas, there are connections also from the primary auditory fields (Plakke and Romanski 2014). This is reflected in the fact that in some neurons in the frontal cortex exhibit responses to auditory features as well as auditory task-related responses and auditory working memory (Romanski and Goldman-

Rakic 2002). It has been documented, in humans and other species, that the frontal lobe is involved in auditory detection and discrimination (Poremba et al., 2004; Zatorre, Bouffard, and Belin 2004). Recently, it has been shown that orbitofrontal neurons directly alter the auditory processing of A1 neurons in the awake mice at the earliest stage of sound processing via fast top-down modulations (Winkowski et al. 2018). The recurrent connections between auditory and frontal areas, henceforth, represent an interacting circuit involved in rapid task-dependent adaptation of auditory information flow and corresponding behaviors (Fritz et al. 2010).

We now include a short paragraph about the recurrent anatomical and physiological connection between primary and secondary auditory and frontal cortices, as a potential source of the findings revealed by our study on page 9, 2nd column, which now states the following:

“Such influence is based on the recurrent circuitry between auditory cortex and higher order top-down regions, e.g. parietal and frontal areas^{4,43–48}. This might reflect a neuronal basis for auditory response properties of frontal lobe neurons⁴⁹, its involvement in auditory detection and discrimination^{50,51} and fast top-down response modulation of A1 neurons during behavior via the frontal cortex⁵².”

Furthermore, we now follow the reviewer’s suggestion and included the data of time windows 500-1000ms into the supplement of the study. Find according changes in the revised version of the manuscript on page 7, 1st column, end 1st paragraph:

“In addition, we analyzed a time window of 500-1000 ms after each stimulus presentation (stimulus duration: 200 ms), in order to separate the relative modulation of cortical layer activity by sensory-driven effects from the task-related, but potentially temporally distributed information. This analysis revealed a similar pattern of cortical activity being modulated by the choice accuracy, which is hence present also independently from the stimulus-dominated auditory response (cf. Supplementary Fig. 6).”

Minor comments:

Some sentences read a bit long occasionally, and therefore give the impression of missing commas that could improve intelligibility.

We proofread the entire manuscript and split any sentence longer than 30 words into two.

Introduction:

Last sentence: what do you mean by “instructive mediator”?

We wanted to emphasize the integrative circuit function of the ACx. As the sentence states this already clearly, the statement of an “instructive mediator” was tautologous. We omitted it in the revised version without any change of the intention of the authors.

Methods:

To simplify the task of a reader not familiar with NeuroNexus nomenclature, please mention the number of channels of the probe and the spacing between them.

We now state the explicit configuration of the electrodes on page 2, 2nd column, 1st paragraph.

Why was the frequency range different between A1 frequency tuning characterization in anaesthetized and awake animals (i.e. 0.25-32 and 0.25-16 kHz)?

Page 2, 2. Column was a typo. We presented frequencies always spanning over 7 octaves from 0.25 kHz – 16 kHz.

Sentence before eq. 1: "...inter-trial (ITS)..” is there a word missing? Shuttles? (the acronym seems to be redefined further ahead in “Data analysis”).

Typo corrected.

Subsection “Acoustic stimuli”: sentence “Two electrostatic...” seems incomplete. Sometimes the tense of some verbs is present. Consider changing those to past tense. (e.g. first sentence, “Data analysis” subsection).

We would like to thank the reviewer for their careful reading and proofread the entire manuscript again.

“Multichannel recordings during training”, last sentence: “leave” for “left”?

Corrected.

CSD analysis section, second paragraph, third sentence? “is” missing?

Corrected.

“Mixed-effects logistic regression”: Missing period mark before the second appearance of GLMM. (right after ref. 33).

Corrected.

Results:

Fig. 2B: insets are a little bit unaligned with each other.

Corrected.

“Auditory cortex represents...”, third sentence: change “report” for “reported”.

Corrected.

Sentence starting with “Holm-corrected...”: should “value” here be plural?

Corrected

Subsection “Representation of contingency...”, 3rd paragraph: change “was also predicting” for “predicted”.

Corrected.

Discussion:

Subsection “Correlates of motor...”, 2nd sentence: remove the comma after “hypothesize”.

Corrected.

Response to Reviewer 2

The study by Zempeltzi et al., uses laminar CSD in auditory cortex of freely moving gerbils to understand if and how task-related variables are encoded across layers. The study is extremely carefully executed, and uses an elegant generalized linear-mixed effects model to interpret the results. The results are important for the field by carefully showing the enriched encoding of choice in supergranular layers during discrimination.

I have no major concerns.

Reply to Reviewer #2:

We would like to thank the reviewer for the careful reading and the appreciation of our study. Please find changes according to the fruitful comments and suggestions raised by the reviewer in our revised manuscript. We are convinced that the updated version of the manuscript increased in clarity and rationale of our applied procedures.

Minor Concerns:

d' learning curves for the data in Figure 1B, right, would be valuable information.

We now added the d' learning curves as a Supplementary Figure 3 accompanying Figure 1b. For generating d' values for the detection phase, we have used the hit rates of both CS+ and contrasted it against the rate of spontaneous inter-trial shuttles (cf. Happel et al., 2014).

Much of the analyzes focus on comparing e.g. physiological parameters on hits to CRs, or FAs to misses. It would be helpful to do full 2-way ANOVAs on all four conditions.

The reviewer refers to Figure 3a from the initial submission here. We decided to use one-way rmANOVAs for individual tone presentations, as we were interested if the 4 different cases derived from the two stimuli (go and nogo) and the two possible behavioral outcomes (jump and no jump), would differ at each time point. Hence, a posthoc analysis of these differences contains 6 comparisons for each time point. In our initial analysis based on a one-way rmANOVA, posthoc comparisons of these differences contained 6 levels. However, we see the argument of the reviewer to test the entire data using a two-way rmANOVA for the corresponding data set with the main

factors outcome (hit/miss/false alarm/correct rejection) and tone order. Correspondingly with the reviewers idea, we found significant effects for the main factors outcome and tone order, as well as for their interaction in both the detection and discrimination phase: Detection: outcome $F_{3,24} = 40.63$ ($p < 0.001$, $\eta_{gen}^2 = 0.66$), tone order $F_{3,96} = 26.58$ ($p < 0.001$, $\eta_{gen}^2 = 0.25$), interaction $F_{9,96} = 10.05$ ($p < 0.001$, $\eta_{gen}^2 = 0.27$); Discrimination: outcome $F_{3,21} = 31.67$ ($p < 0.001$, $\eta_{gen}^2 = 0.68$), tone order $F_{3,84} = 26.39$ ($p < 0.001$, $\eta_{gen}^2 = 0.23$), interaction $F_{9,84} = 17.52$ ($p < 0.001$, $\eta_{gen}^2 = 0.38$).

For a post-hoc analysis of a full 2-way rmANOVA, we could in theory compare between 16 cases overall – if we decided to investigate the interaction between outcome and tone order in detail this would lead to 120 possible comparisons between factor levels in total. However, as we were mainly interested in comparing how cortical activity differed between the four outcomes (hit/miss/false alarm/correct rejection), we would like to use the strong interaction effects of the two-way rmANOVA with the behavioral outcomes and the time axis to motivate a subsequent testing of the time bins in separation. We therefore restrict the posthoc comparisons on the individual time bins in accordance with the previous analysis. This would lead to 24 instead of 6 posthoc comparisons and hence yield a more conservative corrected significance level compared to our previous analysis. We found that most effects reported in the initial manuscript still hold significance, although all significant differences during the first tone presentation vanished (which conforms to our hypothesis that the first tone in a trial is only mildly modulated by the subsequent behavioral choice). The interpretation of the study therefore remained unchanged and was even strengthened. We believe that this new statistical analysis is a valuable improvement which we will now incorporate into the revised manuscript.

Please find a revised text in the results section on page 6, 2nd column:

“We further differentiated how the cortical recruitment depends on the decision made by the animal. We compared z-normalized AVREC RMS values during a 500 ms window beginning with each CS onset. In a first step, we tested the dependence of the animals’ choice options and the time bins throughout the trial, by a two-way rmANOVA with the main factors outcome and tone order (Fig. 3). We found significant effects for both main factors, outcome and tone order, as well as their interaction in both the detection and discrimination phase (Detection: outcome $F_{3,24} = 40.63$ ($p < 0.001$, $\eta_{gen}^2 = 0.66$), tone order $F_{3,96} = 26.58$ ($p < 0.001$, $\eta_{gen}^2 = 0.25$), interaction $F_{9,96} = 10.05$ ($p < 0.001$, $\eta_{gen}^2 = 0.27$); Discrimination: outcome $F_{3,21} = 31.67$ ($p < 0.001$, $\eta_{gen}^2 = 0.68$), tone order $F_{3,84} = 26.39$ ($p < 0.001$, $\eta_{gen}^2 = 0.23$), interaction $F_{9,84} = 17.52$ ($p < 0.001$, $\eta_{gen}^2 = 0.38$). In order to test the differences of the behavioral outcomes at each time point separately, we used restricted Holm-corrected posthoc comparisons at each of the CS+ presentations.”

The y axes in Figure 2B are not sufficiently labeled with tick marks. Labels in Figure 5B and 6B of what each of the 3 ‘cylinders’ corresponds to would be helpful.

We have labeled the right plot in Figure 2b and included a statement in the Figure captions 5b and 6b.

The gray traces in 5A and 6A are too light to see. They’re line weight should be increased.

Done

More justification should be presented to support that they are recording truly in A1 (as opposed to AAF or a belt region).

We now included data from the recordings during the implantation in this response for the reviewers inspection. Pure-tone derived tuning widths on the level of LFP/CSD measures differ between awake and anesthetized recordings. We have recently published a study on a preprint server (currently under peer review), where we compared CSD frequency tunings in the awake and under ketamine anesthesia from animals of the same data set (Deane et al, 2019). Please find below the peak amplitude tuning curves of the dominant layer III/IV sink measured under ketamine anesthesia (during implantation of the recording electrodes) and the awake condition (Figure R3, *left*). The comparison reveals the stronger frequency tuning under ketamine anesthesia compared to awake recordings.

In order to further reveal the implantation of the electrode within the primary auditory cortex, mean onset latencies are a commonly used indicator reflecting early onset activity relayed via the subcortical auditory pathway. We did not observe significant differences in the onset latency of the granular sink in layer III/IV between both conditions (Figure R1, *right*). For stimulation with the best frequency, onset latencies were in the range of around 18ms. This is typically reported for CSD-derived onset latencies obtained from recordings in the field A1 in primary auditory cortex of the Mongolian gerbil (cf. Happel et al., 2010). We also now reference the study by Deane et al. (2019) in the revised version and explain the influence of the anesthesia status on the CSD-derived frequency tuning in A1 within the caption of the corresponding Supplemental Figure 4.

Figure R3. Tuning curves of the mean peak amplitude and the mean onset latencies of the dominant sink in thalamocortical recipient layers III/IV between the awake and anesthetized group. *Left*, Mean peak amplitude tuning curves (centered to the best frequency; BF) showed stronger tuning during ketamine anesthesia compared to awake recordings due to a rampant recurrent excitation (cf. Deane et al., 2019). This is reflected by significant main effects of a two-way rmANOVA for group (G) and frequency (F), as well as a significant interaction (X) of both factors (indicated in the figure panel). *Right*, We did not observe significant differences of the earliest onset latency of the dominant layer III/IV sink between both groups. Stable onset latencies for BF stimulation in the range of ~18ms in both conditions matches the parameters reported before for the field A1 of primary auditory cortex in the Mongolian gerbil (cf. Happel et al., 2010, Happel and Ohl, 2017). Correspondingly, an rmANOVA only reported significant effects for the factor frequency, but not for group or their interaction. Figures adapted from Deane et al., (2019; currently in revision).

Accordingly, we now state on page 3, 1st column, 2nd par:

“Based on the measurements, we observed a rather flat frequency tuning of the dominant early synaptic inputs in the passively listening gerbil before and after each training phase (detection and discrimination) indicating that tonotopic plastic changes are not the main source of our findings (Supplementary Fig. 4).”

Response to Reviewer 3

This study elegantly show using CSD analysis in awake behaving mice performing a detection or a discrimination task in a shuttle box, that task-related decisions strongly participate in the recruitment of cortical activity, and that in some way it is possible to predict the decision of the animal based on CSD responses.

The data is sound and valuable, and although we know already that task parameters such as reward or engagement are encoded in auditory cortex, the fact there is some information about the decisions of the animal is new. That said, we do not know here whether this information is contained in the spiking activity of auditory cortex neurons, as CSD is mainly reflecting synaptic activity, including extrinsic inputs, such as feedback. Despite the clarity of the study and of the manuscript, the analysis and presentation of the results is very compact and could be extended.

Reply to Reviewer #3:

We would like to thank the reviewer for the careful reading and the appreciation of our study. Please find changes according to the fruitful comments and suggestions raised by the reviewer in our revised manuscript. Specifically, we performed a classifier approach (question 1) and a new analysis on the temporal time course of our effects (question 3), which we suggest to include as supplementary information and into the main manuscript, respectively. We are convinced that the updated version of the manuscript now better enunciates the layer-specificity of our findings and resolve their contribution also on a temporal level.

Specific Comments:

Prediction of decisions (or sound identity) is only based on GLMs applied to the 500ms before an escape or an absence of escape decision, showing that decision is a predicting factor of the CSD. It would be very informative to train classifiers and measure what is the fraction of decision that can be correctly predicted.

The reviewer raises an indeed interesting question here. In our initial manuscript, we reported the modulation of the tone-evoked response preceding a behavioral decision within a time window of 500ms (tone duration 200ms; cf. Fig. 3). In order to test the relationship of a dependent variable of cortical layer-specific activity with the behavioral outcome of the animal, we decided to perform statistical modeling and implemented a GLMM (Figs. 4-6). The corresponding R^2_m value of each GLMM model allowed us to evaluate how well the respective synaptic activity can explain the variation in the behavioral outcomes of the animals. With this approach, we did not aim to maximize the accuracy of our prediction in respect to the behavioral outcomes, but rather to investigate the underlying circuit mechanisms of the relationship between the physiological data and behavioral outcomes by utilizing statistical models. These models allow us to evaluate the usefulness of different physiological variables in this prediction (with the R^2_m), while accounting for within-animal variability.

Nevertheless, we recognize that classification techniques have been highly influential in the field. Our data set generally allows to determine how accurate one can classify the physiological data with respect to a certain task variable (e.g. frequency) or behavioral outcome. Henceforth, we have implemented a linear support vector machine (SVM) classifiers, which was trained for targets reflecting either stimulus-related aspects of auditory processing or processing of task-dependent information (see Figure R4 below). Individual SVM-based classification models were trained for bimodal targets reflecting either the stimulus-frequency (1 kHz /4 kHz), the behavioral choice of the animal (Go/NoGo), or the correctness of the choice option. We applied these individual SVM-based

classifiers (linear kernel; nested k-fold cross-validation of classifiers) to the entire CSD matrix in order to reveal class separation based on the columnar spatiotemporal response pattern. We chose time bins of 50ms before tone presentation (baseline) and directly after tone presentation to feed into the classifier. Further, we built our class separation on balanced accuracy, as most targets were not balanced (i.e. correct trials vs incorrect trials and jump vs. no jump). The balanced accuracy captures accuracies with unbalanced data better than regular accuracy measures. We found that SVM-based class separation was significantly above chance level (50%) on all tested classes of stimulus frequency, the correctness of choice and the conditioned response during the detection or discrimination behavior. We further tested class separation of the stimulus frequency before tone onset (baseline), in order to verify the 50% criterion of the SVM classifier. Best accuracies were derived for the class 'stimulus', which corresponded to the 1 kHz or 4 kHz tone yielding averaged accuracies across subjects of 78-82%. Please note, while during detection both tones were Go-stimuli, the two stimuli had opposing contingencies during the discrimination phase. Correspondingly, the classification of the target "correctness" during the detection phase separated trials with a conditioned response or not (hit vs. miss, respectively; accuracy = ~64%). During discrimination, correctness was related to separate hits and correct rejection trials from false alarms and miss trials (accuracy = ~70%). A classifier trained on separating trials in which the animal showed a conditioned response or not during the discrimination phase was less accurate than on the target correctness during detection and discrimination and revealed significantly lower accuracies (accuracy = ~61%).

We think that the reviewer's question about classification approaches could be shared by other readers and generally serves the purpose to use classification approaches in order to generate hypotheses, which need further substantiation by other quantitative approaches (cf. Vu et al., 2018). We further think that our approach of statistical modelling is indeed doing so and it is tailored to our intended research question. We therefore would like to propose to include the new SVM analysis in the new version of the manuscript as a new Supplementary information (see also new Supplementary Fig. 5). In our view, the results are in line with the findings of the initial GLMM approach showing differential contributions of cortical activity in dependence of motor initiation, choice accuracy, etc. We see that the results accompany our initial findings without substantially broadening the impact and would hence rather present this as a supplementary information. We are open to discuss this strategy with the reviewer and editor.

Please find a reference to this new classification analysis on page 6, 2nd column:

"Additionally, we could reveal that two-dimensional CSD data measured in our experiment generally allows to qualitatively dissociate activity patterns utilizing a support vector machine classifier approach (Supplementary Fig. 5). In order to quantify existing differences of the overall columnar activity strength, we compared the root mean square values of the AVREC (AVREC RMS; z-normalized) calculated for the entire trace in each trial (Fig. 2b)."

Figure R4. Class separation based on linear SVM classifiers (n=8). Balanced accuracies (n=8; \pm s.e.m.) are plotted for stimulus frequency, correctness of responses and conditioned responses (CR). Note that correctness and conditioned response during the detection phase is the same (as a CR is always referred to as a hit). In all cases, we found the class separations to be significantly higher than 50% chance level (basel.). Highest accuracies were found for stimulus separation, which yielded \sim 80% accuracy. Bars indicate significant differences tested based on Holm-corrected levels of significance tested by posthoc Student's t-tests.

The GLM measure hides a little the differences between layers that the author often mention. From CSD examples, it seems that all layers are up modulated by Go decisions, with small differences in the degree of modulation. To better appreciate the level of cross-layer difference, the authors could plot the difference between data and a homogenous modulation model.

We opted for GLMM as a method for statistical modeling. We modeled either the entire columnar response (AVREC RMS) or the responses from individual layers. We believe that this is indeed a strength of our statistical analysis approach, as we can interpret the R^2 m given by our GLMM model as an effect size. We now included a better explanation of the interpretation of the R^2 m of the GLMM as an effect size to be able to compare layer-specific results. By comparing the effects sizes of the individual models, we can make assumptions about the concrete contributions of individual cortical layers to the animals' behavioral choices. These findings were reflected in the schematic parts of Figures 4, 5 and 6 indicated by shaded circles reflecting the effect size. However, we agree with the reviewer that this point was not made explicitly clear. We henceforth now adapted the shades to a continuous scaling to better highlight the differences across layers and state this also in the caption of the revised Figures 4-6 (and see also our new Figure 7 according to the next point of the reviewer).

According to the effect size interpretation for the rmANOVA tests, we have adapted a more comprehensible and corresponding explanation of the interpretation of the R^2 m in the methods section on page 5, 2nd column. We now explicitly state ranges of the R^2 m for small, medium and large effect sizes following the same rationale as for the η_{gen}^2 . Please see corresponding edits on page 5, 1st column and 2nd column:

"The generalized eta squared η_{gen}^2 is reported as measure of effect size calculated using the R package DescTools^{31,32}. In general, we interpret effect sizes to be small for $\eta_{gen}^2 \leq 0.1$, medium for $0.1 < \eta_{gen}^2 < 0.25$, and large for $\eta_{gen}^2 \geq 0.25$ ³²."

and

“The R^2m represents the variance in the dependent behavioral variable (on the logistic link scale) explained by the fixed effect of the respective CSD variable (across subjects), while the R^2c reflects the total variance explained by the model’s fixed and random effects, respectively³⁸. In a binary GLMM, the R^2m is independent of sample size and dimensionless, which allows comparing fits across different datasets³⁸. An R^2m of 0.2 thus means that 20% of the variance in the binary outcome can be explained by the cortical activity variable, which was used as the model’s predictor. If the corresponding R^2c is 0.35, the whole model explains 35% of the variance, meaning that an additional 15% of the variance in the outcome can be explained by the variability between animals. The R^2m can hence be used to estimate the effect size, which we did in accordance with the η_{gen}^2 from rmANOVA tests and report small effects for $R^2m \leq 0.1$, medium effects for $0.1 < R^2m < 0.25$, and large effects for $R^2m \geq 0.25$ ³⁵.”

Very little is said about timing. How far back can you predict a decision. Maybe using classifiers, the authors should provide a curve quantifying the progression of the prediction over time until and eventually after the decision of the animal.

The reviewer raised a very relevant question that we are able to answer with our experimental design. In the previous version of the manuscript, we show how cortical responses were modulated over the trial length (Figure 3), but not in dependence of the final decision of the animal.

We have now designed a new analysis, where we, in accordance with our previous approach and with respect to the reviewer’s idea, modeled GLMMs in a time-resolved manner referenced to the reaction time of an animal. Briefly, we found R^2m values to generally increase over the trial duration until a behavioral choice was made, also pronouncing layer-specific differences. While during detection, infragranular layers allowed the prediction of the animal’s choice even at 2-3 CS+ presentations before the actual response, layers III/IV and I/II did so only moderately at the last CS+ presentation preceding the reaction. During discrimination, activity in cortical layers I/II and III/IV, however, allowed us to correctly predict the occurrence of correct rejections of up to 3 CS-presentations before the animal’s reaction. The R^2m -effect sizes relative to the behavioral choices revealed that behavioral outcomes can be increasingly well predicted over the trial duration, which is in agreement with the concept of sensory-based evidence accumulation in order to guide decision making (e.g. Bizley et al., 2013; Tsunada et al., 2015). We believe that this new analysis significantly increased the current study and helps to also better enunciate the specific roles of the individual cortical layers in an auditory decision making task. We henceforth included it as a new Figure 7 in the revised version of the manuscript.

Accordingly, we have also included a new section in the results on page 8, 1st column:

“**Accumulating evidence of task-related information across auditory cortical layers.** In order to reveal the temporal accumulation of such task-dependent information during a trial, we further analyzed time-resolved R^2m values for behavioral choices, which generally increased over the trial duration preceding an animal’s reaction (Fig. 7). While infragranular layers Va-VI showed moderate R^2m values even 2-3 CS+ presentations before the actual response (up to 3-4.5 seconds), layers III/IV and I/II only allowed moderate predictions of the animal’s choice. During discrimination, activity in cortical layers I/II and III/IV, however, allowed us to correctly predict the occurrence of correct rejections of up to 3 CS-presentations before the animal’s reaction, which is particularly pronounced in contrast to false alarm trials. Largest effects were found when comparing hit vs. correct rejections revealing the accumulating evidence of choice accuracy over the trial duration.”

This led to further changes of the abstract and at several locations in the discussion. See for instance on page 8, 2nd column:

“Time-resolved analysis of cortical activity allowed to measure these representations to emerge as accumulating evidence over the trial duration in agreement with the concept of accumulating evidence in sensory cortex.”

And further on page 10, 2nd column:

“Overall, our results show that the layer-specific population activity in sensory cortex is highly dependent on the task design and reflects the performance of the individual subject in agreement with an accumulating sensory evidence in auditory cortex ultimately leading to a behavioral choice^{8,64}.”

References

- Amy Poremba. 2004. Species-specific calls evoke asymmetric activity in the monkey's temporal poles. *Nature* **427**:445–448. doi:/doi.org/10.1038/nature02268
- Bakeman R. 2005. Recommended Effect Size Statistic. *Behav Res Methods* **37**:379–384. doi:10.3758/BF03192707
- Bizley JK, Cohen YE. 2013. The what, where and how of auditory-object perception. *Nat Rev Neurosci* **14**:693–707. doi:10.1038/nrn3565
- Deane KD, Brunk, MGK, Curran AW, Zempeltzi MM, Ma J, Lin X, Abela F, Aksit S, Deliano M, Ohl FW, Happel MFK. 2019. Ketamine anesthesia induces gain enhancement via recurrent excitation in granular input layers of the auditory cortex. bioRxiv. doi: https://doi.org/10.1101/810978
- Fritz JB, David S V., Radtke-Schuller S, Yin P, Shamma SA. 2010. Adaptive, behaviorally gated, persistent encoding of task-relevant auditory information in ferret frontal cortex. *Nat Neurosci* **13**:1011–1019. doi:10.1038/nn.2598
- Happel MFK, Deliano M, Handschuh J, Ohl FW. 2014. Dopamine-modulated recurrent corticoefferent feedback in primary sensory cortex promotes detection of behaviorally relevant stimuli. *J Neurosci* **34**:1234–47. doi:10.1523/JNEUROSCI.1990-13.2014
- Happel MFK, Jeschke M, Ohl FW. 2010. Spectral integration in primary auditory cortex attributable to temporally precise convergence of thalamocortical and intracortical input. *J Neurosci* **30**:11114–27. doi:10.1523/JNEUROSCI.0689-10.2010
- Happel MFK, Ohl FW. 2017. Compensating Level-Dependent Frequency Representation in Auditory Cortex by Synaptic Integration of Corticocortical Input. *PLoS One* **12**:e0169461. doi:10.1371/journal.pone.0169461
- Nanda P, Morris A, Kelemen J, Yang J, Wiest MC. 2020. Evoked frontal and parietal field potential signatures of target detection and response inhibition in rats performing an equiprobable auditory go/no-go task. *eNeuro* **7**:1–18. doi:10.1523/ENEURO.0055-19.2019
- Petrides M, Pandya DN. 1988. Association fiber pathways to the frontal cortex from the superior temporal region in the rhesus monkey. *J Comp Neurol* **273**:52–66. doi:10.1002/cne.902730106
- Plakke B, Romanski LM. 2014. Auditory connections and functions of prefrontal cortex. *Front Neurosci* **8**:1–13. doi:10.3389/fnins.2014.00199
- Romanski LM, Goldman-Rakic PS. 2002. An auditory domain in primate prefrontal cortex. *Nat Neurosci* **5**:15–16. doi:10.1038/nn781
- Sheikhattar A, Miran S, Liu J, Fritz JB, Shamma SA, Kanold PO, Babadi B. 2018. Extracting neuronal functional network dynamics via adaptive Granger causality analysis. *Proc Natl Acad Sci U S A* **115**:E3869–E3878. doi:10.1073/pnas.1718154115
- Smith JL, Johnstone SJ, Barry RJ. 2008. Movement-related potentials in the Go/NoGo task: The P3 reflects both cognitive and motor inhibition. *Clin Neurophysiol* **119**:704–714. doi:10.1016/j.clinph.2007.11.042
- Tsunada J, Liu ASK, Gold JI, Cohen YE. 2015. Causal contribution of primate auditory cortex to auditory perceptual decision-making. *Nat Neurosci* **19**:135–142. doi:10.1038/nn.4195
- Vu MAT, Adalı T, Ba D, Buzsáki G, Carlson D, Heller K, Liston C, Rudin C, Sohal VS, Widge AS, Mayberg HS, Sapiro G, Dzirasa K. 2018. A shared vision for machine learning in neuroscience. *J Neurosci* **38**:1601–1607. doi:10.1523/JNEUROSCI.0508-17.2018
- Winkowski DE, Nagode DA, Donaldson KJ, Yin P, Shamma SA, Fritz JB, Kanold PO. 2018. Orbitofrontal Cortex Neurons Respond to Sound and Activate Primary Auditory Cortex Neurons. *Cereb Cortex* **28**:868–879. doi:10.1093/cercor/bhw409
- Zatorre RJ, Bouffard M, Belin P. 2004. Sensitivity to Auditory Object Features in Human Temporal Neocortex. *J Neurosci* **24**:3637–3642. doi:10.1523/JNEUROSCI.5458-03.2004

REVIEWERS' COMMENTS:

Reviewer #1 (Remarks to the Author):

The authors made an outstanding job with the revised paper. I enjoyed greatly reading the new version and the response letter to the reviewers. In particular, the new Figure 7 showing the build-up of evidence for behavioural choices along a trial duration is indeed very interesting. I have no major points to raise. However, a few minor remarks could help the authors when proof-reading their paper in the future. Changes for these suggestions I leave at the authors' discretion.

Methods:

Page 2, 2nd par.: "from a loudspeaker (Tannoy arena satellite KI-8710-32) in 1 m distance to the animal". "at" instead of "in"?

Page 2, last paragraph, 2nd line: "electrically shielded and sound-proof chamber". Would "sound-proofed" work better?

Results

Page 7, last paragraph: "...the 500 ms windows around the tone presentation". I still don't understand if this is centred at the tone somehow, or if it is after the tone. I assume while reading that this should be after tone presentation. If this is so, perhaps the authors could make it clearer by substituting "around", for the benefits of other readers like myself.

Figure 4, caption: "The boxplots above and below the curves represent the mean (bar), interquartile range (box) the full range of data (whiskers)". "and" missing after "(box)"?

The data presented in the study are very compelling. Therefore, I agree with the authors that it might be more comprehensive for readers to visualize a schematic of the AVREC traces, and I understand in principle their decision of keeping them. However, I still think that whenever possible the actual data should be shown, if only for the sake of transparency. The AVREC depicted in the new Supplementary Figure 3 is qualitatively very similar to the schematics in the main Figures, and the insets in the same Supplement. Moreover, it seems to me that the visualization of these data is also very similar to the schematics. I believe that the paper would profit largely from showing these traces (even just the mean and without the s.e.m., for clarity) instead of schematic curves. Perhaps the AVREC traces could be accommodated to the size and format of the insets that are already in the figures, without altering their complexity too much (i.e. substituting the schematic traces with the real ones?). I am convinced of the main results of the study, as they are robustly backed by statistical analysis and data. Adding the traces would bring yet another layer of strength to the manuscript. Nevertheless, based on what I have discussed above, I feel comfortable leaving this changes at the authors' discretion.

Reviewer #3 (Remarks to the Author):

The authors satisfactorily addressed my comments and the manuscript is, to me, ready for publication.

Response Letter for COMMSBIO-19-1888

Manuscript: Task rule and choice are reflected by layer-specific processing in rodent auditory cortical microcircuits

Please find below the point-by-point responses (in blue) to each comment of the reviewers (in black).

Response to Reviewer 1

The authors made an outstanding job with the revised paper. I enjoyed greatly reading the new version and the response letter to the reviewers. In particular, the new Figure 7 showing the build-up of evidence for behavioural choices along a trial duration is indeed very interesting. I have no major points to raise. However, a few minor remarks could help the authors when proof-reading their paper in the future. Changes for these suggestions I leave at the authors' discretion.

Reply to Reviewer #1:

We would like to again thank the reviewer for the careful reading and the appreciation of our study. Please find changes according to the fruitful comments and suggestions raised by the reviewer in our revised manuscript.

Methods:

Page 2, 2nd paragraph: “from a loudspeaker (Tannoy arena satellite KI-8710-32) in 1 m distance to the animal”. “at” instead of “in”?

We did the corresponding change.

Page 2, last paragraph, 2nd line: “electrically shielded and sound-proof chamber”. Would “sound-proofed” work better?

We did the corresponding change.

Results

Page 7, last paragraph: “...the 500 ms windows around the tone presentation”. I still don't understand if this is centred at the tone somehow, or if it is after the tone. I assume while reading that this should be after tone presentation. If this is so, perhaps the authors could make it clearer by substituting “around”, for the benefits of other readers like myself.

We substituted ‘around’ with the following: ‘500 ms windows beginning at the onset of the tone presentation’ and hope that it now gained clarity.

Figure 4, caption: “The boxplots above and below the curves represent the mean (bar), interquartile range (box) the full range of data (whiskers)”. “and” missing after “(box)”?

We did the corresponding change.

The data presented in the study are very compelling. Therefore, I agree with the authors that it might be more comprehensive for readers to visualize a schematic of the AVREC traces, and I understand in principle their decision of keeping them. However, I still think that whenever possible the actual data should be shown, if only for the sake of transparency. The AVREC depicted in the new Supplementary Figure 3 is qualitatively very similar to the schematics in the main Figures, and the insets in the same Supplement. Moreover, it seems to me that the visualization of these data is also very similar to the schematics. I believe that the paper would profit largely from showing these traces (even just the mean and without the s.e.m., for clarity) instead of schematic curves. Perhaps the AVREC traces could be accommodated to the size and format of the insets that are already in the figures, without altering their complexity too much (i.e. substituting the schematic traces with the real ones?). I am convinced of the main results of the study, as they are robustly backed by statistical analysis and data. Adding the traces would bring yet another layer of strength to the manuscript. Nevertheless, based on what I have discussed above, I feel comfortable leaving this changes at the authors' discretion.

We now moved the raw traces from the Supplemental Figure 3 into the main Figure 2 as subpanel b. We thereby combine both, namely showing the raw traces, as suggested by the reviewer, and also keeping the comprehensive schematics in now panel 2c, that the reviewer also appreciated.